# miR-1199-5p and Zeb1 function in a double-negative feedback loop potentially coordinating EMT and tumour metastasis

Maren Diepenbruck[1], Stefanie Tiede[1], Meera Saxena[1], Robert Ivanek [1], Ravi Kiran Reddy Kalathur[1], Fabiana Lüönd[1], Nathalie Meyer-Schaller[1] & Gerhard Christofori[1]

Epithelial tumour cells can gain invasive and metastatic capabilities by undergoing an epithelial–mesenchymal transition. Transcriptional regulators and post-transcriptional effectors like microRNAs orchestrate this process of high cellular plasticity and its malignant consequences. Here, using microRNA sequencing in a time-resolved manner and functional validation, we have identified microRNAs that are critical for the regulation of an epithelial–mesenchymal transition and of mesenchymal tumour cell migration. We report that miR-1199-5p is downregulated in its expression during an epithelial–mesenchymal transition, while its forced expression prevents an epithelial–mesenchymal transition, tumour cell migration and invasion in vitro, and lung metastasis in vivo. Mechanistically, miR-1199-5p acts in a reciprocal double-negative feedback loop with the epithelial–mesenchymal transition transcription factor Zeb1. This function resembles the activities of miR-200 family members, guardians of an epithelial cell phenotype. However, miR-1199-5p and miR-200 family members share only six target genes, indicating that, besides regulating Zeb1 expression, they exert distinct functions during an epithelial–mesenchymal transition.

[1] Department of Biomedicine, University of Basel, 4058 Basel, Switzerland. Nathalie Meyer-Schaller and Gerhard Christofori contributed equally to this work. Correspondence and requests for materials should be addressed to N.M-S. (email: nathalie.meyer-schaller@unibas.ch) or to G.C. (email: gerhard.christofori@unibas.ch)

An epithelial-to-mesenchymal transition (EMT) as well as its reversal, a mesenchymal-to-epithelial transition (MET), reflect two gradual, well-controlled processes during embryogenesis and wound healing in adults to promote tissue and organ formation and homeostasis. Both processes induce a global reorganization of a cell's constitution and allow switching back and forth between two different cell phenotypes to endow the necessity of tissue plasticity[1]. In the context of malignant tumour progression, epithelial tumour cells can undergo an EMT upon diverse extracellular stimuli and consequently gain metastatic capabilities. Among many growth factors and environmental cues, such as tissue hypoxia, transforming growth factor β (TGFβ) strongly activates the dedifferentiation process in epithelial tumour cells and, thus, induces global changes in a cell's transcriptional and post-transcriptional networks[2–4]. This allows tumour cells to disseminate from the primary tumour and to intravasate and survive in the blood circulation. At the distant organ, cells extravasate into the organ parenchyma and eventually grow out as lethal metastases, possibly promoted by a MET[5–8]. An EMT also provides cells with increased chemoresistance, thus impeding efficient therapy of malignant mesenchymal cancer cells[9, 10].

MicroRNAs (miRNAs) represent a class of ~22 nucleotide-long non-coding RNAs that can regulate gene expression at the post-transcriptional level by either inducing target messenger RNA (mRNA) degradation or preventing mRNA translation[11–13]. In the context of an EMT, miRNA-200 family members (miR-200a/b/c, miR-141 and miR-429) take a central stage: they are required to maintain an epithelial cell morphology by degrading the transcripts of the EMT-inducing transcription factors (TFs) Zeb1 and Zeb2[14–17]. Members of the miR-200 family bind to specific seed sequences in the 3′ untranslated region (3′ UTR) of Zeb1 and 2 mRNAs and destabilize them. During an EMT, members of the miR-200 family are downregulated in their expression, which results in the increased expression of Zeb1 and 2. Conversely, Zeb1 and Zeb2 directly suppress the transcription of miR-200 family members[14–17]. Such a double-negative feedback loop is a major example for a reciprocal TF-miRNA regulation during an EMT. Similar other molecular switches also regulate EMT/MET plasticity and malignant tumour progression[18–22].

Here, we report the identification of miR-1199-5p, as a repressor of EMT, tumour cell invasion and metastasis, which comparable to miR-200 family members targets Zeb1 mRNA for degradation. Conversely, Zeb1 represses the expression of miR-1199-5p and of the miR-200 family. However, miR-200 family members and miR-1199-5p seem to exert distinct functions; they share only six of their many target mRNAs, among them Zeb1.

## Results

### Identification of EMT-associated miRNAs

To identify regulatory miRNAs involved in the gradual process of an EMT, we performed miRNA sequencing on a detailed time course of a TGFβ-induced EMT in normal murine mammary gland cells (NMuMG subclone E9; NMuMG/E9). Analysis of the kinetics of miRNA transcript regulation during an EMT in a time-resolved manner identified 32 differentially expressed miRNAs. Unsupervised hierarchical clustering illustrated that approximately half of the differentially expressed miRNAs showed a continuous increase in their expression during an EMT, whereas the other half exhibited decreased expression (Fig. 1a). In order to identify the miRNAs functionally impacting on a TGFβ-induced EMT, we performed a microscopy-based screen in which NMuMG/E9 cells were transfected with miRNA mimics and cultured in the absence or presence of TGFβ for 4 days (Fig. 1b). Subsequently, mesenchymal cell characteristics were monitored by high-content fluorescence microscopy analysis and quantified, including the deposition of the extracellular matrix protein fibronectin and the formation of focal adhesions and of actin stress fibres[23]. Nine out of 32 differentially expressed miRNAs were able to block or at least delay the EMT process: miR-125b-5p, miR-181b-2-3p, miR-1247-3p, miR-200a-3p, miR-200b-3p, miR-429-3p, miR-1199-5p, miR-145a-3p and miR-504-5p. Conversely, the forced expression of miR-145a-5p and miR-6944-3p promoted an EMT in epithelial NMuMG/E9 cells. Changes in cell morphology, the localization of the epithelial adhesion junction protein E-cadherin as well as the mRNA levels of E-cadherin, N-cadherin, fibronectin and Zeb1 further confirmed the impact of the different miRNAs on the EMT process (Supplementary Fig. 1a–c).

Increased cell migration is a functional output of EMT, which allows tumour cell invasion and dissemination into the surrounding tissue[1]. In a second screening step, we identified those miRNAs that were able to affect mesenchymal tumour cell migration (Fig. 1c). Mesenchymal, highly migratory and tumorigenic Py2T cells that have been treated >20 days with TGFβ[24] were transiently transfected with mimics of EMT-affecting miRNAs. Four out of the 11 miRNAs tested significantly reduced cell migration in a trans-well Boyden chamber assay: miR-200b-3p, miR-429-3p, miR-1199-5p, and to a lesser extent miR-125b-5p (Supplementary Fig. 2a, b). As a result, our experimental strategy identified miRNAs whose transcriptional regulation and function affected both a TGFβ-induced EMT and mesenchymal breast cancer cell migration (Fig. 1d). Four miRNAs significantly fulfilled these criteria: the miRNA-200 family members miR-200b-3p and miR-429-3p, and miR-125b-5p and miR-1199-5p. Since ectopic expression of miR-1199-5p, an EMT-regulatory miRNA, seemed to affect EMT with similar potency as the well-studied miR-200 family members[15, 17, 18], we further investigated the functional role and the mechanisms of action of miR-1199-5p in the regulation of an EMT.

### miRNA-1199-5p inhibits EMT and tumour cell invasion

The continuous downregulation of miR-1199-5p expression during a TGFβ-induced EMT was due to transcriptional repression as determined by a miR1199-promoter/luciferase-reporter assay during a TGFβ-induced EMT in NMuMG/E9 cells and Py2T murine breast cancer cells, and in mesenchymal Py2T cells and metastatic 4T1 murine breast cancer cells treated with TGFβ for >20 days (Fig. 2a, Supplementary Fig. 3a, b). Analysis of miR-1199 expression in different human breast cancer cell lines[25] further confirmed a higher expression in epithelial than in mesenchymal breast cancer cells (Supplementary Fig. 3c).

The forced expression of miR-1199-5p by the transfection of a construct encoding a 1199-5p miRNA mimic caused sustained epithelial cell morphology in NMuMG/E9 and in human untransformed mammary gland MCF10A cells induced to undergo an EMT by TGFβ treatment (Fig. 2b–e, Supplementary Fig. 3d–f). The cells maintained their epithelial morphology and the cell surface localization of the adherens junction protein E-cadherin and of the tight junction protein ZO-1 at the plasma membrane, while miR-Ctr-transfected cells lost these epithelial characteristics and progressed with an EMT (Fig. 2b, Supplementary Fig. 3d). The reorganization of cortical actin to stress fibres as well as the formation of focal adhesions were also prevented by miR-1199-5p mimics (Fig. 2c). Mesenchymal markers, such as fibronectin, vimentin or N-cadherin were repressed at the protein (Fig. 2d, Supplementary Fig. 3e) and mRNA (Fig. 2e, Supplementary Fig. 3f) expression levels in miR-1199-5p-transfected cells. Furthermore, miR-1199-5p induced a significant decrease in Zeb1 mRNA, however, transcript levels of other key EMT TFs, such as Zeb2[26] or Sox4[27] remained unchanged (Fig. 2e).

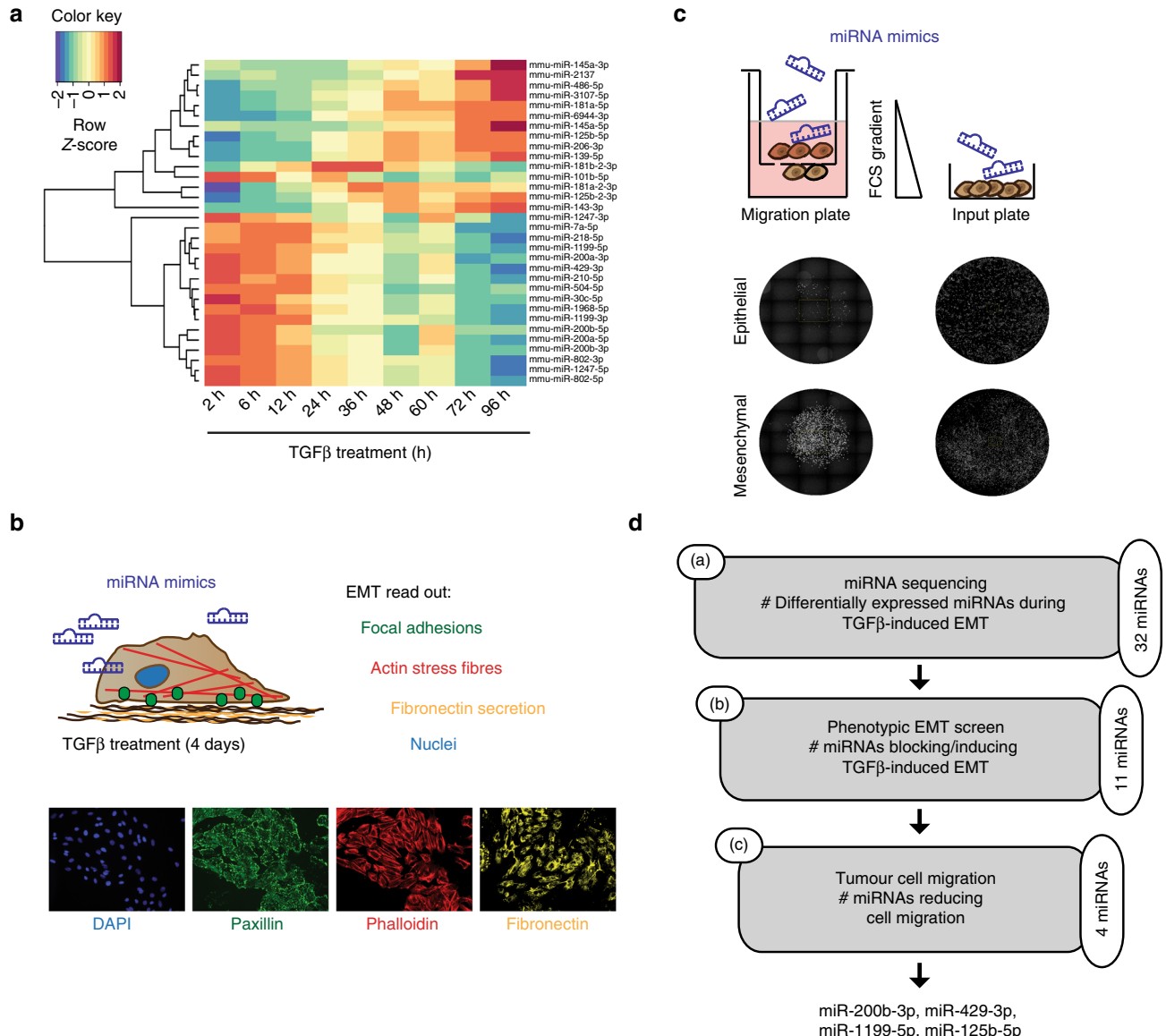

**Fig. 1** Identification of miRNAs critical for an EMT and cell migration. **a** Expression profiling of miRNAs during a TGFβ-induced EMT in NMuMG/E9 cells. NMuMG/E9 cells were treated with TGFβ for the time points indicated, and total RNA was isolated for miRNA sequencing. The heat map summarizes the hierarchical clustering of differentially expressed miRNAs (log2FC(±2); FDR <0.05) compared to epithelial, untreated cells during an EMT according to the indicated colour scale. **b** Identification of miRNAs controlling EMT. NMuMG/E9 cells were transfected with miRNA mimics for individual miRNAs and analysed for mesenchymal characteristics in the absence and presence of TGFβ for 4 days. Formation of focal adhesions (green), actin cytoskeleton reorganization to actin stress fibres (red), fibronectin deposition (yellow) and nuclei (blue) were visualized by high-content fluorescence screening microscopy. **c** Identification of miRNAs regulating mesenchymal tumour cell migration. Mesenchymal, migratory Py2T cells (>20 days TGFβ) were transfected with different miRNA mimics, plated in 96-well Boyden chamber migration inserts within a FCS gradient and in parallel on a 96-well input plate. Cell nuclei were imaged using a fluorescence screening microscope and quantified. Migrated cells were normalized to the total cell number on the input plate. **d** Scheme depicting the workflow and the number of (**a**) differentially expressed miRNAs during an EMT, (**b**) the number of identified miRNAs functionally contributing to a TGFβ-induced EMT and (**c**) the number and names of the miRNAs functionally contributing to mesenchymal tumour cell migration

A gain in cell migration and invasion can be one of the consequences of a TGFβ-induced EMT and allows tumour cells to leave the primary tumour and intravasate into the blood circulation and to extravasate at a distant organ[1, 5]. Ectopic expression of miR-1199-5p displayed only minor effects on the early stages of an EMT in Py2T cells (Supplementary Fig. 3g–i), yet it induced a significant reduction in migration and invasion of mesenchymal (>20 days TGFβ) Py2T cells, as analysed by trans-well Boyden chamber assays (Fig. 2f). The inhibitory effects of miR-1199-5p on in vitro cell migration and invasion were also observed in 4T1 cells treated for >20 days with TGFβ (Fig. 2g). In summary, miR-1199-5p is sufficient to sustain an epithelial cell phenotype.

**The direct target genes of miR-1199-5p during an EMT.** To elucidate the genome-wide function of miR-1199-5p during an EMT, we performed RNA-sequencing analysis on NMuMG/E9 cells transiently transfected with either a miR-1199-5p mimic or a miR-Ctr mimic and cultured for 4 days in the presence of TGFβ.

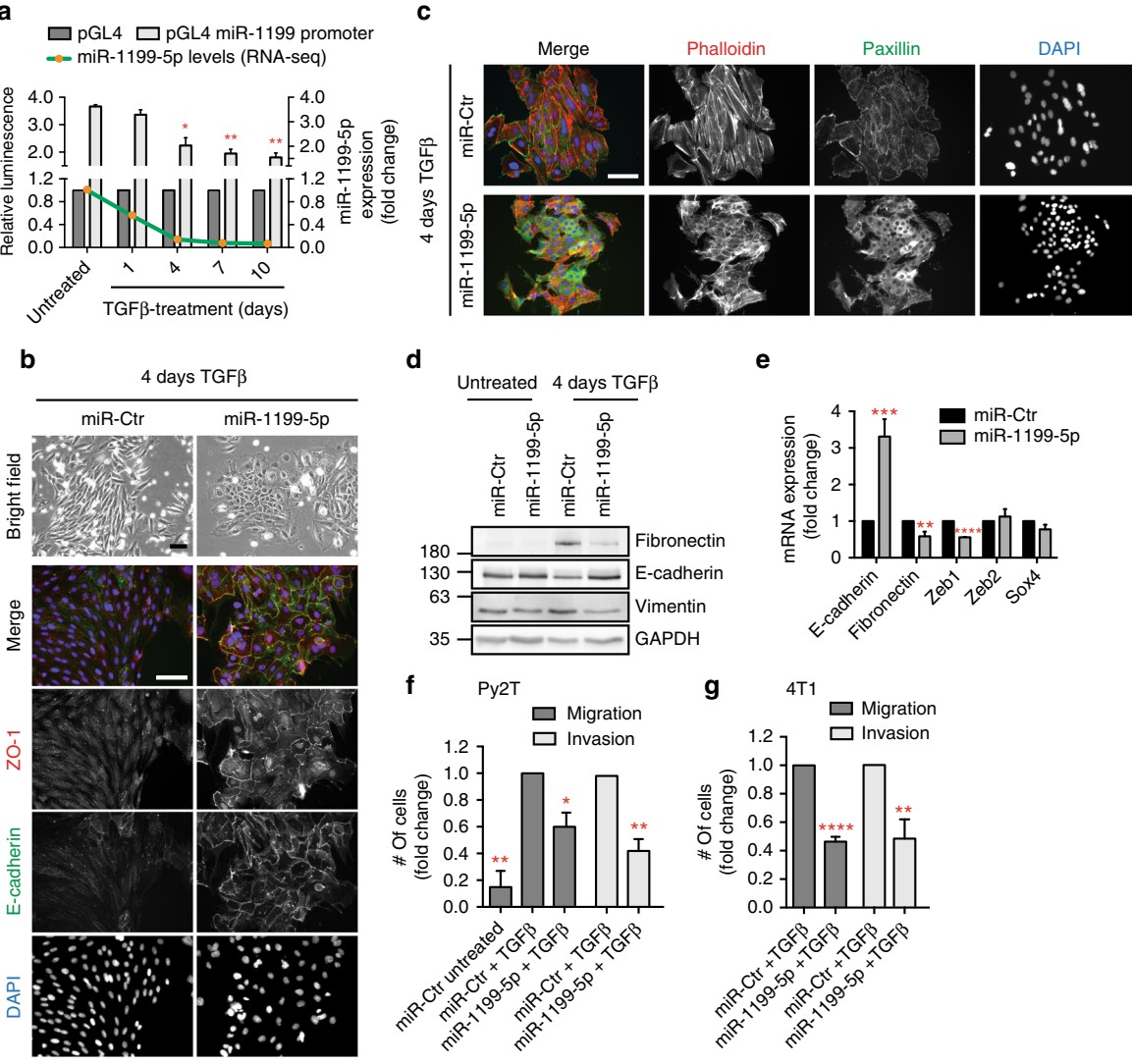

**Fig. 2** miRNA-1199-5p inhibits an EMT and tumour cell migration and invasion. **a** miR-1199-5p transcript and miR-1199 promoter activity levels decrease during a TGFβ-induced EMT. Green line: expression profile of miR-1199-5p during EMT in NMuMG/E9 cells as determined by RNA sequencing. Grey bars: miR-1199 promoter activity during an EMT. NMuMG/E9 cells were treated with TGFβ for the time points indicated and transfected with a Renilla luciferase reporter along with either a miR-1199 promoter Firefly luciferase reporter (pGL4 miR-1199 promoter) or a control reporter (pGL4; Supplementary Fig. 3b). Relative luminescence (Firefly/Renilla) was calculated and normalized to the control reporter (mean fold changes ± s.e.m.; n = 3; significance determined by an unpaired, two-sided t test; *P < 0.05, **P < 0.01). **b, c** NMuMG/E9 cells were transiently transfected with the miRNA mimics indicated and cultured in presence of TGFβ for 4 days (4-day TGFβ). Bright-field images of NMuMG/E9 cells illustrate the differences in cell morphology. Immunofluorescence images of NMuMG/E9 cells visualize different epithelial and mesenchymal cell structures as indicated. Scale bars: 100 μm. **d, e** Immunoblot (**d**) and quantitative RT-PCR mRNA (**e**) expression analysis of NMuMG/E9 cells transiently transfected with miRNA mimics for the EMT markers indicated (mean fold changes ± s.e.m.; n = 6; significance determined by an unpaired, two-sided t test, Student's t test; **P < 0.01, ***P < 0.001, ****P < 0.0001). **f, g** Mesenchymal (>20 days TGFβ) Py2T (**f**) and 4T1 (**g**) cells were transiently transfected with the miRNA mimics indicated and re-plated within a FCS gradient of a Boyden chamber migration or invasion insert. Nuclei of transmigrated and invaded cells were quantified after 18 h (mean fold changes ± s.e. m.; migration: n = 3; invasion Py2T: n = 3, 4T1: n = 4; significance determined by an unpaired, two-sided t test; *P < 0.05, **P < 0.01, ****P < 0.0001)

As expected, forced expression of miR-1199-5p induced a block in a TGFβ-induced EMT in these cells and led to an overall anti-correlative (r = −0.324) transcriptomic profile compared to genes regulated in their expression during an EMT (miR-Ctr 4d vs. miR-Ctr 0d) (Fig. 3a). We found 787 genes differentially expressed (log2 fold change of ±1 (log2FC(±1)); False Discovery Rate (FDR) <0.05) by the ectopic expression of miR-1199-5p during an EMT. Subsequent functional annotation analysis by DAVID (Database for Annotation, Visualization and Integrated Discovery)[28, 29] for biological processes (GO (gene ontology)) and cellular pathways (Kyoto Encyclopedia of Genes and Genomes; KEGG) revealed their involvement predominantly in cell adhesion processes, extracellular matrix (ECM)-receptor inter-actions and focal adhesions (Fig. 3b).

In order to delineate the mechanism by which miR-1199-5p maintains an epithelial cell morphology, we set out to determine its direct mRNA targets during an EMT. Computational analysis by miRWalk[30] revealed 1789 potential target mRNAs of miR-1199-5p, which display a seed sequence in their 3′ UTR, 5′ UTR or coding sequence (CDS). Overlaying these mRNA targets with genes regulated by miR-1199-5p during an EMT (Fig. 3a; 787 genes) uncovered 90 target genes of which 66 displayed a significant reduction in their transcript levels upon the forced expression of miR-1199-5p (Fig. 3c, Supplementary Table 1).

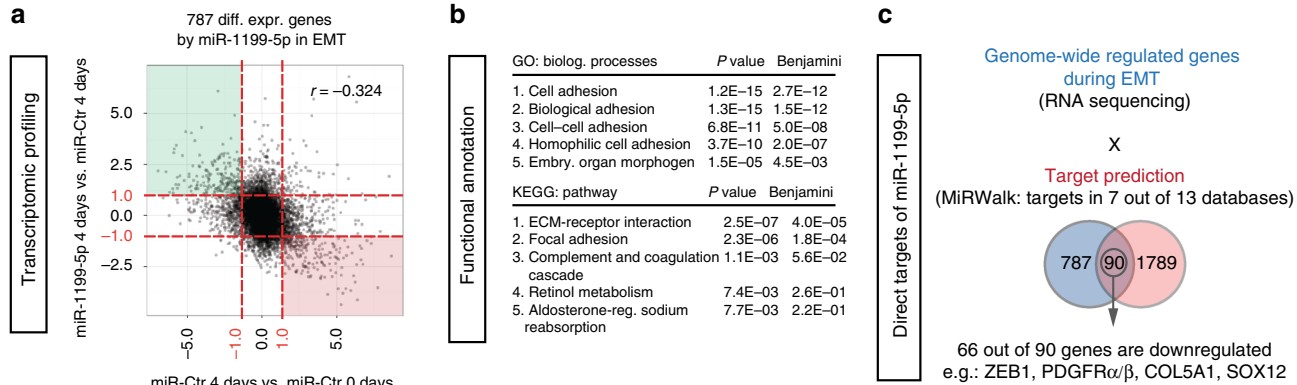

**Fig. 3** Targets of miR-1199-5p function during an EMT. **a** Overall gene expression regulation by miR-1199-5p during a TGFβ-induced EMT. RNA-sequencing analysis was performed on NMuMG/E9 cells transiently transfected with a miR-1199-5p or a negative control (miR-Ctr) mimic. Cells were cultured for 4 days in the presence (4 days; miR-Ctr- and miR-1199-5p-transfected cells) or absence of TGFβ (only miR-Ctr-transfected cells). The scatterplot depicts overall gene expression (log2FC) anti-correlation between miR-1199-5p 4 days vs. miR-Ctr 4 days over miR-Ctr 4 days vs. miR-Ctr 0 days. Differential expression analysis (red dashed line: log2FC(±1); False Discovery Rate (FDR) <0.05) identified 787 genes regulated by miR-1199-5p during an EMT. Green quadrant: increased gene expression; red quadrant: decreased gene expression. **b** Functional annotation clustering analysis of differentially expressed genes from **a**. Gene ontology (GO) and Kyoto Encyclopedia of Genes and Genomes (KEGG) pathway analyses by DAVID for the top five biological processes (top), pathways (bottom) and their associated *P* and Benjamini–Hochberg values. **c** Identification of miR-1199-5p direct targets in EMT. The Venn diagram depicts the number of genes differentially expressed upon miR-1199-5p expression during an EMT (blue), predicted direct targets of miR-1199-5p by miRWalk2.0 (red) and the number of overlapping genes. Sixty-six out of 90 genes demonstrate a downregulation in their expression upon forced expressed of miR-1199-5p in EMT, including the EMT TF Zeb1

Among these genes, we identified the key EMT TF Zeb1[31] with a potential conserved 8-mer seed sequence for miR-1199-5p in its 3′ UTR (Fig. 4a).

**miR-1199-5p directly targets Zeb1 mRNA.** The zinc-finger E-box-binding homeobox TF Zeb1 has a well-established role as transcriptional repressor of epithelial genes and, hence, as an activator of an EMT[3, 31]. Zeb1 function is associated with increased stemness, cell survival and metastasis[32, 33], and high levels of Zeb1 have been linked to aggressive breast cancer subtypes, therapy resistance, high risk for distant metastasis and poor survival[34, 35]. As expected, Zeb1 transcript levels were increased during a TGFβ-induced EMT of different murine and human cellular systems (Supplementary Fig. 4a). Furthermore, small-interfering RNA-mediated knockdown of Zeb1 maintained an epithelial cell morphology in NMuMG/E9 and MCF10A cells induced to undergo an EMT and significantly reduced the migratory properties of mesenchymal (>20 days TGFβ) Py2T and 4T1 cells in trans-well Boyden chamber assays (Supplementary Fig. 4b–e).

We next examined whether miR-1199-5p indeed regulated Zeb1 levels during an EMT. Ectopic expression of miR-1199-5p in NMuMG/E9 and Py2T cells induced to undergo an EMT resulted in the stabilization of cell junction protein E-cadherin, while nuclear levels of Zeb1 were reduced (Fig. 4b, Supplementary Fig. 4f). Immunoblotting analyses for Zeb1 confirmed the repression of Zeb1 expression by miR-1199-5p in these murine cells (Fig. 4c, Supplementary Fig. 4g). Even though the seed match of human miR-1199-5p is not perfectly complementary to the seed sequence in the Zeb1 3′ UTR (mouse and human miR-1199-5p seed matches differ in one base), forced expression of hsa-miR-1199-5p still significantly reduced Zeb1 mRNA (Supplementary Fig. 3f) and protein (Supplementary Fig. 4h) levels in human MCF10A cells. These results further imply that even imperfect binding of a miRNA to its target RNA can efficiently downregulate its expression.

To validate the direct regulation of Zeb1 by miR-1199-5p on the post-transcriptional level during an EMT we made use of two Zeb1 3′ UTR luciferase reporter constructs, one containing the species conserved, wild-type miR-1199-5p seed sequence and the other one carrying a mutated version of the seed sequence with five nucleotides exchanged (Fig. 4a, d). Transient transfection of a miR-1199-5p mimic and the Zeb1 3′ UTR wild-type reporters in NMuMG/E9 cells revealed a significant decrease in luminescence, which was not observed with the mutant version of the reporter (Fig. 4d). Together, these data identify the key EMT TF Zeb1 as a direct target of miR-1199-5p, which thus represses Zeb1 expression at the post-transcriptional level and prevents an EMT.

**Zeb1 directly controls the expression of miR-1199-5p.** An important mechanism that appears to be responsible for EMT cell plasticity are double-negative feedback loops between miRNAs and key EMT TFs, functioning as molecular switches for various cell differentiation states[18–20, 22, 36, 37]. Because miR-1199-5p regulates the expression of Zeb1, we assessed whether Zeb1 would in turn regulate the expression of miR-1199-5p during an EMT. siRNA-mediated ablation of Zeb1 expression in NMuMG/E9 cells cultured in the presence of TGFβ for 4 days and transfected with either a miR-1199-promoter luciferase-reporter (pGL4 miR-1199 promoter) or a control promoter luciferase-reporter (pGL4; Supplementary Fig. 3b) revealed a significant increase in miR-1199 promoter activity upon loss of Zeb1 (Fig. 5a, left). Conversely, transient overexpression of Myc-tagged Zeb1 in epithelial NMuMG/E9 cells significantly decreased miR-1199 promoter activity (Fig. 5a, right). Furthermore, the regulation of miR-1199 promoter activity by loss or gain of function experiments of Zeb1 also correlated with an increase or decrease in endogenous miR-1199-5p transcript levels, respectively (Fig. 5b).

The gene encoding for miR-1199-5p is located within the first CDS of an unknown, protein-coding gene (2210011C24Rik) on chromosome 8 in the mouse genome. Its localization is conserved in the human genome, where it is located in the first CDS of LOC113230, an orthologue of 2210011C24Rik, on chromosome 19 (Supplementary Fig. 5a). Notably, the transcript levels of the murine host gene are significantly downregulated during a TGFβ-induced EMT in different cellular models, as it was observed for

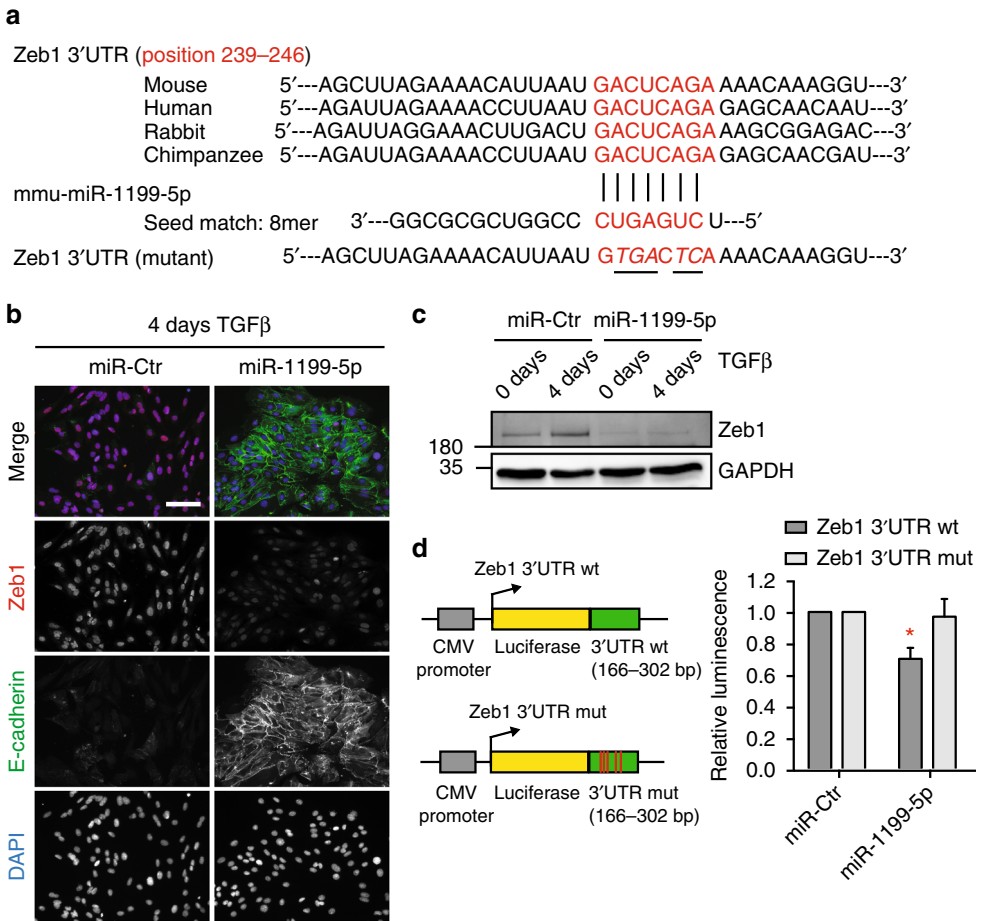

**Fig. 4** miR-1199-5p directly controls Zeb1 expression. **a** Species-conserved miR-1199-5p-binding site in the 3′ UTR of Zeb1. The scheme represents: top: sequence alignments of predicted binding sites of miR-1199-5p (red; mouse: position 239–246) in Zeb1 3′ UTRs of different species. Middle: alignment of the 8mer seed match in mouse miR-1199-5p. Bottom: mutated miR-1199-5p seed sequence in the 3′ UTR of Zeb1 mRNA utilized for the control reporter construct (Zeb1 3′ UTR mut) in **d**. Exchanged nucleotides are underlined. **b**, **c** Forced expression of miR-1199-5p reduces Zeb1 nuclear localization and protein levels during an EMT. NMuMG/E9 cells were transiently transfected with miRNA mimics as indicated and cultured in the absence (0 day) or presence of TGFβ (4 days). Immunofluorescence (**b**) and immunoblotting (**c**) analyses illustrate the differences in Zeb1 protein levels. Scale bar: 100 μm. **d** Post-transcriptional regulation of Zeb1 by miR-1199-5p. NMuMG/E9 cells were transfected with the miRNA mimics indicated, with a Renilla luciferase reporter construct and with either a Zeb1 3′ UTR wild-type (wt) or a Zeb1 3′ UTR mutant (mut) Firefly luciferase reporter construct. Relative luminescence (Firefly/Renilla) was calculated and normalized to miR-Ctr-transfected cells (mean fold changes±s.e.m.; $n = 3$; significance determined by an unpaired, two-sided $t$ test; *$P < 0.05$)

miR-1199-5p (Supplementary Fig. 5b; Fig. 2a). Furthermore, gain and loss of function studies by siRNA-mediated ablation or transient overexpression of Zeb1 in NMuMG/E9 cells led to an increase or decrease in 2210011C24Rik expression, respectively (Supplementary Fig. 5c, d), indicating a co-regulation of miR-1199-5p and its host gene.

The murine miR-1199/2210011C24Rik promoter region encompasses several E-box motifs (CANNTG; N = G or C) as potential Zeb1-binding sites (Fig. 5c). We next performed chromatin immunoprecipitation (ChIP) for endogenous Zeb1 in epithelial Py2T cells and in 4 days TGFβ-treated Py2T cells, a time point displaying a robust increase in Zeb1 expression (Supplementary Fig. 4a), followed by quantitative RT-PCR analysis with various primer pairs covering different regions of the miR-1199/2210011C24Rik promoter. These experiments confirmed a direct binding of Zeb1 to a region with the closest proximity (+51/−52 bps) to the transcriptional start site (TSS) (Fig. 5d). Notably, Zeb1 binding was only observed in cells undergoing an EMT and not in their epithelial counterparts (Fig. 5d). Using the miR-1199/2210011C24Rik promoter luciferase reporter (Supplementary Fig. 3b), we individually mutated

each of four Zeb1 E-box-binding sites (CAGGTG to CATTTG). Only mutation of E-box 1 (−18 bp from TSS) significantly ablated Zeb1-mediated repression of luciferase activity in NMuMG/E9 cells (Fig. 5e).

Together, the results show that Zeb1 is a direct transcriptional repressor of the *miR-1199* gene. Hence, Zeb1 and miR-1199-5p act in a reciprocal fashion to repress each other.

**Comparing miR-1199-5p and miR-200s function during an EMT**. MiR-1199-5p's function during an EMT as well as its regulation by the EMT TF Zeb1 resembles the activities of members of the miR-200 family, two of which, miR-200b-3p and miR-429-3p, have been identified by our functional EMT screen (Fig. 1)[14–17].

Comparable to miR-1199-5p, miR-200b-3p and miR-429-3p transcript levels are also strongly reduced during a TGFβ-induced EMT in NMuMG/E9 cells (Supplementary Fig. 6a, Fig. 2a). The forced expression of miR-200b-3p or miR-429-3p in human MCF10A (Supplementary Fig. 6b, top) and murine NMuMG/E9 cells (Supplementary Fig. 1a–c) also prevented an EMT and

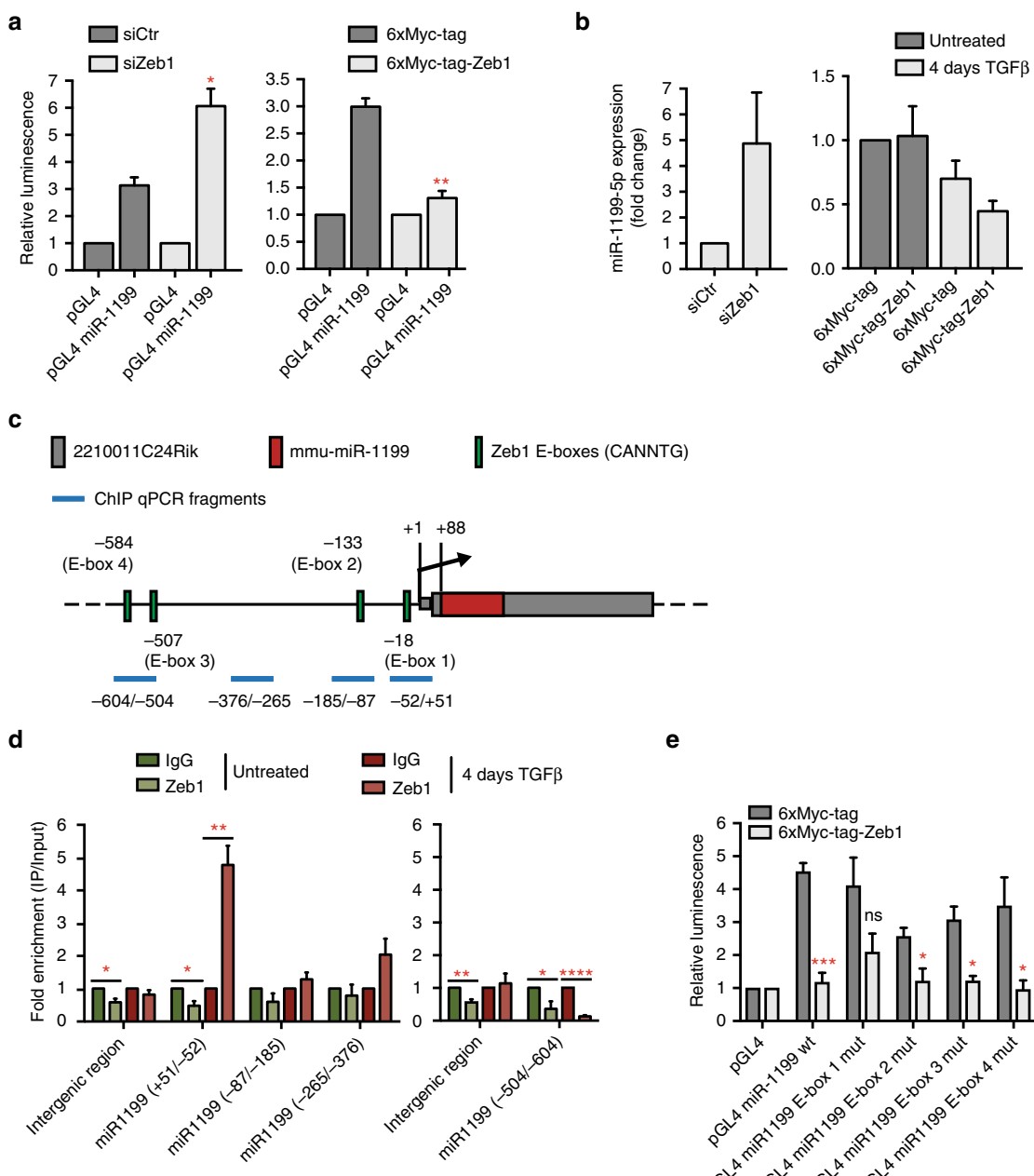

**Fig. 5** Zeb1 directly regulates miR-1199-5p expression. **a** Zeb1 controls the promoter activity of the miR-1199 gene. NMuMG/E9 cells were transfected with a miR-1199 promoter Firefly luciferase reporter construct, a Renilla luciferase reporter construct as well as with (left) a siRNA against Zeb1 (siZeb1) or a negative control siRNA (siCtr) or (right) a Zeb1 expression construct (6xMyc-tag-Zeb1) or a negative control construct (6xMyc-tag). Cells transfected with siRNAs were cultured for 4 days with TGFβ, while cells transfected with 6xMyc or 6xMyc-tagged Zeb1 were cultured in the absence of TGFβ. Relative luminescence (Firefly/Renilla) was calculated and normalized to the control Firefly luciferase reporter (pGL4; mean fold changes±s.e.m.; $n = 3$; significance determined by an unpaired, two-sided $t$ test; *$P < 0.05$, **$P < 0.01$). **b** Zeb1 controls miR-1199-5p transcript levels. NMuMG/E9 cells were transfected with siRNAs (left) and Zeb1 constructs (right) as described in **a** and further cultured in the absence or presence of TGFβ. MiR-1199-5p transcript levels were examined by RT-PCR analysis (mean fold changes±s.e.m.; left: $n = 5$; right: $n = 3$). **c** Schematic presentation of the genomic localization of the murine miR-1199 gene and its promoter region. Red: mmu-miR-1199; grey: 2210011C24Rik gene; green: E-boxes (CANNTG, $N = $ G or C); blue: promoter fragments examined by ChIP-qPCR analysis. **d** Zeb1 directly binds to the miR-1199 promoter. Chromatin of Py2T cells cultured in the absence (green) and presence of TGFβ (red) was subjected to chromatin immunoprecipitation with antibodies against Zeb1 followed by RT-PCR analysis using primers amplifying different regions of the miR-1199 promoter illustrated in **c**. An intergenic region was used as negative control. Data were normalized to control IgG and are presented as mean fold enrichment above background±s.e.m. ($n = 3$; significance determined by an unpaired, two-sided $t$ test; *$P < 0.05$, **$P < 0.01$, ****$P < 0.0001$). **e** Analysis of E-box-binding motifs in the miR-1199 promoter as potential Zeb1-binding sites. NMuMG/E9 cells were transfected and cultured as described in **a** (right). Relative luminescence (Firefly/Renilla) was calculated as described in **a** (mean fold changes±s.e.m.; $n = 3$; significance determined by an unpaired, two-sided $t$ test; *$P < 0.05$, ***$P < 0.001$)

maintained an epithelial morphology in the presence of TGFβ. However, similar to miR-1199-5p, expression of miR-200b-3p and miR-429-3p mimics in tumorigenic Py2T cells failed to completely block mesenchymal cell morphology, even though the mRNA levels of E-cadherin and Zeb1 were significantly increased or decreased, respectively (Supplementary Fig. 6b, c). Also comparable to miR-1199-5p, ectopic expression of miR-200b-3p or miR-429-3p significantly reduced cell migration and invasion of mesenchymal Py2T and 4T1 cells (Supplementary Fig. 2, Supplementary Fig. 6d). Notably, all three miRNAs are mechanistically embedded in a double-negative feedback regulation with Zeb1[18, 19] (Fig. 4, Fig. 5, Supplementary Fig. 6e).

Since miR-200b-3p, miR-429-3p and miR-1199-5p induced comparable functional outputs with regard to an EMT process, we assessed their effects on tumour progression in vivo. Stable expression of miR-1199, miR-200b (which also induced the expression of miR-429 and slightly miR-1199; Supplementary Fig. 6f) and miR-429 in metastatic 4T1 cells tagged by ZsGreen led to a reduction in Boyden chamber trans-well cell migration compared to an empty vector control (Supplementary Fig. 6g) and confirmed our results for their transient overexpression (Fig. 2g, Supplementary Fig. 6d).

Upon orthotopic transplantation of 4T1 cells into the mammary fat pads of immunodeficient NSG and NMRI mice, the forced expression of miR-1199, miR-429 or miR-200b induced a significant reduction in primary tumour growth over time (Fig. 6a). Notably, the high potential of 4T1 cells to metastasize to the lungs was reduced by the expression of miR-1199 and miR-429, however not by the expression of miR-200b (Fig. 6b–d). The failure of miR-200b to repress 4T1 metastasis was surprising, yet has been observed by others as well (G.J. Goodall, personal communication of unpublished data). Examining the metastatic outgrowth of 4T1 cells in the lung, expression of each of the three miRNAs significantly increased the size of the few metastatic nodules compared to the many nodules observed with empty vector-transduced cells (Fig. 6e). Therefore, the repressive growth effect observed for miR-1199, miR-200b and miR-429 in the primary tumour was not maintained in the outgrowth of tumour cells at the metastatic site. Of note, the number of circulating tumour cells isolated from the blood of tumour-bearing mice was decreased by the forced expression of miR-1199 and miR-429, but not by the expression of miR-200b (Fig. 6f, g).

In summary, miR-1199 and miR-429, but not miR-200b, are sufficient to reduce tumour cell intravasation into the blood circulation and the seeding of lung metastases. However, all three miRNAs seem to promote metastatic outgrowth once tumour cells have seeded in the lung parenchyma.

### Common and distinct targets of miR-200s and miR-1199-5p.
The functional similarities between miR-1199-5p and miR-200 family members observed during an EMT in vitro as well as during tumour progression in vivo begs the question about the shared and the distinct functions of miR-200 family members and miR-1199-5p in regulating an EMT. To reveal the transcriptomic effects of the various miRNAs during a TGFβ-induced EMT, we performed RNA sequencing of NMuMG/E9 cells transfected with miRNA mimics for miR-200b-3p, miR-429-3p and miR-1199-5p. As expected, miR-200b-3p or miR-429-3p mimics induced a block in EMT of NMuMG/E9 cells and led to an overall anti-correlative gene expression profile compared to mesenchymal, miR-Ctr-transfected cells ($r = -0.505$ and $r = -0.51$, respectively; Fig. 7a), similar to the profile observed with miR-1199-5p mimic ($r = -0.324$; Fig. 3a). Differential gene expression analysis (log2FC ($\pm 1$); FDR <0.05) revealed 1097 and 1058 genes affected by miR-

200b-3p and miR-429-3p, respectively, during an EMT, of which 982 genes were shared by the two miRNAs (Fig. 7a, b). Since they belong to the same miRNA family, such a high number of co-regulated genes can be explained by an identical seed sequence on their target mRNAs. However, the difference in the effects of the two miRNAs on lung metastasis is surprising (Fig. 6b–d), and we can only speculate whether the small pool of individual target genes for each of the miRNAs is the reason for the different outcomes in vivo.

Directly compared to each other, miR-200b-3p, miR-429-3p and miR-1199-5p share 465 regulated genes during an EMT, which is more than half of the 787 genes regulated by miR-1199-5p alone (Fig. 7b, Fig. 3a). Functional annotation analysis revealed that these genes control the same biological processes and pathways as miR-1199-5p alone, that is, cell adhesion, ECM-receptor interactions and focal adhesions (Fig. 7c, Fig. 3b). A total of 267 genes are exclusively regulated by miR-1199-5p during an EMT.

Target prediction analysis by miRWalk2.0[30] for miR-200b-3p and miR-429-3p together with the differential expression analysis presented in Fig. 7a revealed 54 direct target genes commonly regulated by both miR-200 family members during an EMT (Fig. 7d, Supplementary Table 2), among them the key EMT TFs Snail1, Zeb1 and Zeb2[3]. Six out of the 54 genes were also directly controlled by miR-1199-5p (Fig. 7d). Besides the common target Zeb1, mRNA levels for Ncs1, Cdon, Sox12, Zfp9, Col5a1 were also decreased by all three miRNAs during an EMT (Fig. 7e). Of note, 58 target genes were uniquely regulated by miR-1199-5p.

The results show that miR-1199-5p, miR-200b-3p and miR-429-3p tightly control TGFβ-induced EMT plasticity with similar potency, however, they only share the regulation of six gene transcripts, one of which is the critical EMT TF Zeb1. Yet, each of the miRNAs seems to have distinct functions by repressing a larger number of unique target mRNAs.

### Discussion
A cancer-associated EMT can drive early steps of the metastatic cascade by converting epithelial tumour cells into invasive metastatic cancer cells[1, 5]. MiRNAs are powerful, post-transcriptional regulators of a cell's transcriptome and, therefore, they are able to coordinate changes of a cell's morphology and functional capabilities necessary for an EMT[11–13]. In the present study, we have set out to identify and characterize such EMT-regulatory miRNAs in normal and tumorigenic breast cancer cells. Using miRNA-sequencing analysis of the kinetics of EMT, we identified 32 strongly regulated miRNAs by a global, time-resolved miRNA profile of TGFβ-induced EMT. Subsequently, we have tested the miRNAs' functional contribution to an EMT and to mesenchymal tumour cell migration. Among a number of miRNAs, we identify miR-1199-5p as a strong regulator of an EMT. Its transcript levels as well as its promoter activity are continuously decreased during an EMT. Furthermore, we demonstrate that miR-1199-5p downregulation is required for cell dedifferentiation, for mesenchymal tumour cell migration and invasion, and for lung metastasis formation in vivo. Finally, we report that during an EMT, this exonic miRNA is mechanistically embedded in a reciprocal, direct negative feedback loop with the TF Zeb1. The double-negative feedback loop with Zeb1 during an EMT is reminiscent of the miR-200 family members[14–16, 18, 19]. Indeed, consistent with previous reports, the miR-200 family members miR-200b-3p and miR-429-3p have been identified by our EMT screen as critical mediators of an epithelial cell phenotype[14–17].

While the gain of function approaches of miR-1199-5p and miRNA family members has clearly repressed an EMT, we have

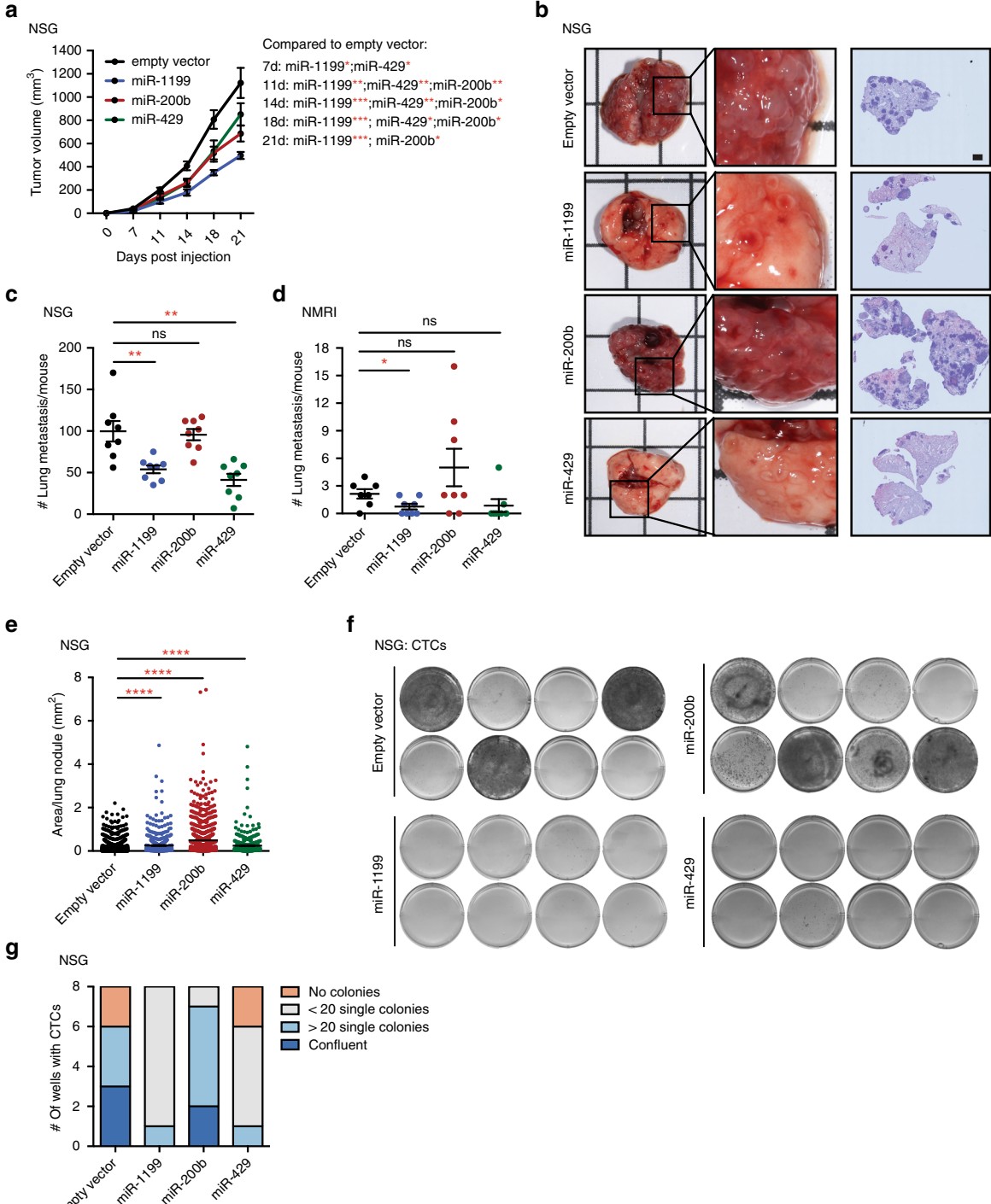

**Fig. 6** miR-1199 and miR-200s repress 4T1 murine breast cancer primary tumour growth and metastasis. **a** ZsGreen-labelled 4T1 cells stably expressing miR-1199, miR-200b or miR-429 were injected into the mammary fat pad of female NSG or NMRI mice as indicated, and primary tumour growth was analysed over time. Statistical significance compared to empty vector control: 7 days: miR-1199*, miR-429*; 11 days: miR-1199**, miR-429**, miR-200b**; 14 days: miR-1199***, miR-429**, miR-200b**; 18 days: miR-1199***, miR-429***, miR-200b*; 21 days: miR-1199***, miR-200b*. **b** Representative bright-field images of whole lungs (left), enlargement of indicated lung areas (middle) and H&E staining of lung sections (right) isolated from mice killed 21 days post cell injection. Scale bar, 1 mm. **c**, **d** The number of lung metastases per mouse was quantified microscopically by H&E staining. **e** The tissue area of all lung nodules was analysed and quantified microscopically by H&E staining. **f**, **g** CTCs isolated from the blood of tumour-bearing mice were isolated 21 days post cell injection and cultured for 5 days ex vivo. Cell colonies were visualized by MTT staining, imaged (**f**) and quantified (**g**) as indicated. NSG: 8 mice per group±s.e.m.; NMRI: 7–8 mice per group±s.e.m.; statistical analyses: Mann–Whitney U test; *P < 0.05, **P < 0.01, ***P < 0.001, ****P < 0.0001

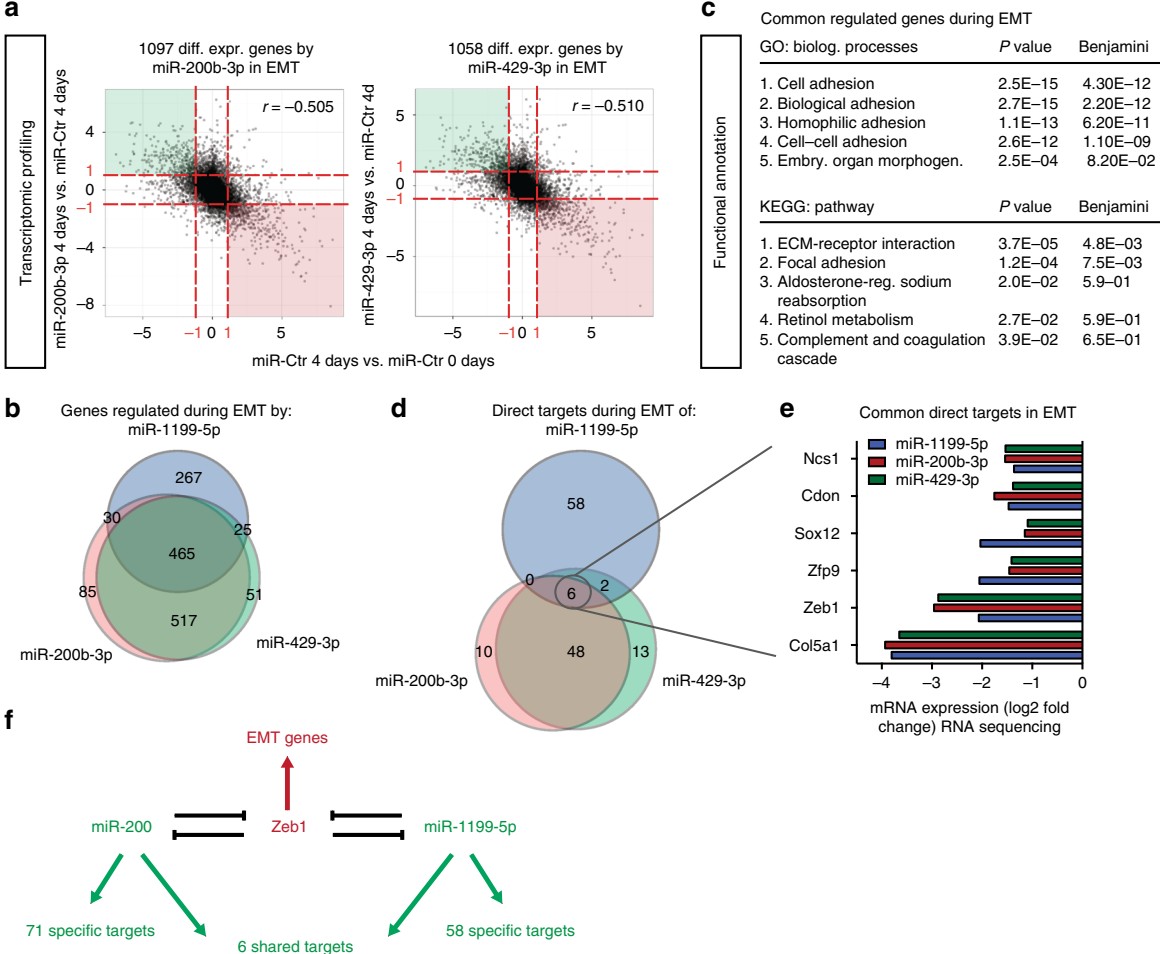

**Fig. 7** Shared and distinct target genes of miR-1199-5p, miR-200b-3p and miR-429-3p. **a** Overall gene expression regulation by miR-200b-3p and miR-429-3p during a TGFβ-induced EMT. RNA-sequencing analyses were performed on NMuMG/E9 cells transiently transfected with a miR-200b-3p, miR-429-3p or a negative control (miR-Ctr) mimic. Cells were then cultured in the presence or absence of TGFβ (0 day vs. 4 days). RNA sequencing and data analysis was performed as described for miR-1199-5p in Fig. 3a. Scatterplots depict the overall gene expression (log2FC) (anti-)correlation between miR-200b-3p 4 days vs. miR-Ctr 4 days (left) or miR-429-3p 4 days vs. miR-Ctr 4 days (right) over miR-Ctr 4 days vs. miR-Ctr 0 day. Differential expression analysis (red dashed line: log2FC(±1); FDR <0.05) identified 1097 genes and 1058 genes regulated during an EMT by miR-200b-3p and miR-429-3p, respectively. **b** The Venn diagram summarizes the number of genes commonly and individually regulated by miR-1199-5p (blue), miR-200b-3p (red) and miR-429-3p (green) during a TGFβ-induced EMT (data from Fig. 7a, Fig. 3a). **c** Functional annotation clustering analysis by DAVID of genes commonly regulated by miR-1199-5p, miR-200b-3p and miR-429-3p during an EMT. Presented are the top five biological processes, pathways (KEGG) and their associated *P* and Benjamini–Hochberg values. **d** The Venn diagram presents the number of commonly and individually regulated, predicted target genes of miR-1199-5p (blue), miR-200b-3p (red) and miR-429-3p (green) during an EMT. MirWalk2.0 was used to predict miRNA target genes that show reduced transcript levels upon forced expression of miR-1199-5p, miR-200b-3p and miR-429-3p during an EMT. **e** Shown are the mRNA levels of genes commonly repressed by miR-1199-5p, miR-200b-3p and miR-429-3p during an EMT (RNA-sequencing data: log2FC compared to miR-Ctr-transfected cells). **f** Schematic representation of the double-negative feedback loops between miR-200 family members and Zeb1 and between miR-1199-5p and Zeb1. Only six target genes are shared between miR-200 and miR-1199-5p among which is Zeb1. Each of them has individual target genes as indicated and thus may affect distinct biological functions

repeatedly failed to demonstrate an effect of loss function approaches on EMT by interfering with miR-1199-5p and also with miR-200 family member activities. Interestingly, transient transfection of a miR-1199-5p sponge construct or a miR-1199-5p inhibitor did not lead to a loss of miR-1199-5p function in epithelial NMuMG/E9 cells, as determined by 3′ UTR reporter assays, yet was not sufficient to increase Zeb1 expression or to induce an EMT. Of note, sponge constructs or miRNA inhibitors against miR-200b-3p and miR-200c-3p also did not alter Zeb1 expression or the induction of an EMT (Supplementary Fig. 7). From these data, we conclude that the regulation of Zeb1 expression on the transcriptional and posttranscriptional level is complex and we suppose that the loss of one Zeb1 regulator

(miR-1199-5p or miR-200b/c-3p) is not sufficient to increase Zeb1 levels and to jumpstart an EMT. The overlapping double-negative feedback loops between miRNAs and TFs may act as a 'buffered system' to compensate for the loss of one regulator and thus ensure a system's functionality.

Our genome-wide analyses of the three miRNAs revealed a large number of regulated genes, which underscores their critical functions during an EMT. However, only six genes were identified as common direct targets of the three miRNAs, including Zeb1, Col5a1, Zfp9, Sox12, Cdon and Ncs1. The latter five are so far 'unknowns' in the context of an EMT and malignant tumour progression, most of which might be of interest considering their potential function. For instance, Sox12, together with Sox11 and

Sox4, is a member of the SoxC TF family, of which Sox4 has an established role in EMT and breast cancer progression[27]. The cell surface receptor encoded by Cdon, similar to the EMT inducer neuronal cell adhesion molecule, consists of immunoglobin/fibronectin type III domains known to mediate cell signalling[38]. Finally, collagen type V alpha 1 (Col5a1) is mainly found in organ tissue together with collagen type I, a major ECM component of breast cancer. Furthermore, Col5a1 facilitates breast cancer formation, invasion and metastasis[39, 40].

In summary, we report the identification and characterization of miR-1199-5p as a miRNA acting in a negative reciprocal regulation with Zeb1 during an EMT. Our findings add this negative feedback loop to other known reciprocal regulations between a miRNA and a key EMT TF[18, 20–22]. These regulatory units may work as functional epithelial/mesenchymal switches supporting the reversibility, robustness and effectiveness of EMT/MET, endowing cells with a high degree of cell plasticity necessary for embryonic development as well as for malignant tumour progression.

## Methods

**Reagents and antibodies.** E-cadherin (BD Transduction Labs, 610182), N-cadherin (Takara, M142), ZO-1 (Zymed, 617300), paxillin (BD, 610052), fibronectin (Sigma-Aldrich, F3648), vimentin (Sigma-Aldrich, V2258), GAPDH (Sigma-Aldrich, G8795), Zeb1 (Cell Signaling, 3396 and Santa Cruz Biotechnology, sc-25388), Alexa-Fluor 488 and 568 (Molecular Probes), secondary horse radish peroxidase (HRP)-conjugated antibodies against mouse and rabbit (Jackson ImmunoResearch), recombinant human TGFβ1 (R&D Systems, 240-B), 4′,6-dia-midino-2-phenylindole (DAPI, Sigma-Aldrich) and phalloidin Alexa-Fluor 568 (Molecular Probes, A12380).

**Cell lines and cell culture.** A subclone of normal murine mammary gland cells (NMuMG/E9)[41], the MCF10A human epithelial mammary gland cell line[42], the Py2T murine breast cancer cell line[24] and the murine metastatic cell line 4T1[43] have been described previously. NMuMG/E9, Py2T and 4T1 cells were grown in Dulbecco's modified eagle medium (DMEM; Sigma-Aldrich) supplemented with foetal calf serum (FCS, 10%; Sigma-Aldrich), glutamine (2 mM; Sigma-Aldrich), penicillin (100 U; Sigma-Aldrich) and streptomycin (0.2 mg/l; Sigma-Aldrich). MCF10A cells were cultured in DMEM/F12 medium supplemented with horse serum (5%; Bioconcept Amimed), insulin (10 μg/ml; Sigma-Aldrich), hydrocortisone (0.5 μg/ml; Sigma-Aldrich), human EGF (0.02 μg/ml; Invitrogen) and cholera toxin (0.01 μg/ml; Sigma-Aldrich). All cell lines were grown at 37 °C, 5% $CO_2$, 95% humidity.

**Plasmids.** pCS3-6xMyc-tag-Zeb1 and pCS3-6xMyc-tag-Zeb2 were a kind gift from T. Brabletz (Friedrich-Alexander-Universität Erlangen-Nürnberg, Germany). The control plasmid pCS3-6xMyc-tag was generated by digesting a pCS3-6xMyc-tag-Zeb2 vector with EcoRI/XbaI, thereby removing the Zeb2 gene but retaining the 6xMyc tag.

*miR-1199 promoter reporter*: The genomic promoter region of miR-1199/2210011C24Rik (808 nucleotides upstream of 2210011C24Rik TSS) of NMuMG/E9 cells was PCR-amplified with the following primers: 5′-tcgactcgaggaccggggaaacactctgta-3′ and 5′-tcgaaagcttgcgtctccatctgcaattccgc-3′, which exhibit restriction enzyme sites for XhoI and HindIII, respectively (underlined). The PCR amplicon was digested with XhoI and HindIII and subcloned into the pGL4.10[luc2] Firefly luciferase reporter vector purchased from Promega.

Using the QuikChange II site-directed mutagenesis Kit (Agilent), four different miR-1199/2210011C24Rik promoter luciferase mutants exhibiting point mutations in the Zeb1 E-box-binding motifs (wt: CAGGTG; mut: CATTTG) at positions −18, −133, −507 and −584 bp from the TSS were generated from the original miR-1199/2210011C24Rik promoter reporter. The following primers were used to introduce the different point mutations:

E-box 1 (−18 bp):
5′-ctcggctcagaggcctccatttgtgtgttttctaccgaag-3′,
5′-cttcggtagaaaacacaaatggaggcctctgagccgag-3′;
E-box 2 (−133 bp):
5′- cagagagccaggtcccgcatttgccagttgagc-3′,
5′-gctcaactggcaaatgcgggacctggctctctg-3′;
E-box 3 (−507 bp):
5′-ccactcgaatccagctggcatttgacgggggcg-3′,
5′-cgccccccgtcaaatgccagctggattcgagtgg-3′;
E-box 4 (−584 bp):
5′-ccttcagtcaacccgtgtgttcatttggcgggaatccg-3′,
5′-cggattcccgccaaatgaacacacggggttgactgaagg-3′.

*Zeb1 3′UTR reporter*: A mouse Zeb1 3′ UTR luciferase reporter for miR-1199-5p was generated by subcloning the murine Zeb1 3′ UTR sequence (166–302 bps) containing a wild-type (wt: GACTCAGA) or mutated miR-1199-5p-binding site (mut: GTGACTCA) into the pMIR-REPORT Firefly luciferase vector (Applied Biosystems) via SpeI/HindIII sites. The wild-type and mutant Zeb1 3′ UTR luciferase reporter for miR-200b-3p and miR-429-3p contain two common miRNA-binding sites (wt: CAGTATTA, CAGTATT; mut: CTCAATAA, GTCTAA) were generated by subcloning the murine Zeb1 3′ UTR sequence (276–431 bps) into the pMIR-REPORT vector as described for Zeb1 3′ UTR/miR-1199-5p. The different DNA fragments were synthesized by Integrated DNA Technologies.

*Plasmids expressing ZsGreen1 and mmu-miR-1199 or mmu-miR-200b or mmu-miR-429*: Stem loop sequences of murine miR-1199, miR-200b and miR-429 plus additional 100 bps up and downstream of the miRNAs were synthesized by Integrated DNA Technologies (miR-1199) or PCR-amplified from genomic DNA of NMuMG/E9 cells with the primers:
miR-200b:
5′-tcgaacgcgtcctccttctgcaatgctctg-3′,
5′-tcgagctagcctaactctttgccccatagcc-3′;
miR-429:
5′-tcgaacgcgtgaagggtgaaccccaagaat-3′ and
5′-tcgagctagccagcggggcctgtatattt-3′.
The miRNA constructs were flanked with MluI and NheI sites (underlined) and subsequently digested and subcloned into the 3′ UTR of ZsGreen of the pmRi-ZsGreen1 vector (Clontech) via the restriction enzyme sites mentioned.

miRNA sponge constructs: constructs with 5× miR-1199-5p, miR-200b/c-3p or scrambled bulged binding sites separated by a spacer sequence (GAATAT) were synthesized and inserted in the 3′ UTR of an enhanced green fluorescent protein reporter gene driven by the CMV promoter (pEGFP-C3, Clontech) by Invitrogen.

**RNA interference.** *miRNA mimic transfection*: NMuMG/E9, MCF10A, Py2T and 4T1 cells were transfected with 20 nM or 50 nM of mouse or human pre-miR miRNA precursors (Ambion) by using Lipofectamine RNAiMax (Invitrogen) according to the manufacturer's instructions. A random sequence has been used as negative control (Ambion, AM17110 control #1). To maintain a miRNA's overexpression, the transfection was repeated every day. All pre-miR miRNA precursors used in this study are listed in Supplementary Table 3.

*miRNA inhibitor transfection*: NMuMG/E9 cells were transfected with 50 nM of miRNA inhibitors against miR-1199-5p from Ambion (miRVana; MH13577) and Exiqon (miRCURY LNA; 4108998).

*siRNA transfection*: 20 nM of siRNA against Zeb1 (Ambion: mouse: s74841, 74843 10 nM each; human: s229971) or a negative control (Ambion: 4390846) was used for transient gene knockdown experiments. Lipofectamine RNAiMax (Invitrogen) was used for the transfection according to the manufacturer's instructions and was repeated every third day.

**Stable miRNA overexpression.** 4T1 cells were serially transfected with pTet-On Advanced and plasmids expressing ZsGreen1 and mmu-miR-1199 or mmu-miR-200b or mmu-miR- 429 or ZsGreen1 only using Lipofectamine 2000 (Invitrogen) and selected with Neomycin and Hygromycin (Clontech). Additionally, cells stably expressing ZsGreen1 were sorted by flow cytometry, and ectopic miRNA expression was validated by quantitative RT-PCR.

**Luciferase reporter assays.** A dual-luciferase reporter assay (Promega) was used to measure miR-1199/2210011C24Rik promoter activity and posttranscriptional repression of Zeb1 3′ UTRs by miR-1199-5p, miR-200b-3p and miR-429-3p according to the manufacturer's instructions. Firefly/Renilla luciferase activity was measured with a luminometer (Berthold Technologies; Centro LB 960).

*miR-1199/2210011C24Rik promoter reporter*: Cells were plated in triplicates in a 24-well plate and treated with TGFβ. Three days before measuring luciferase activity, cells were transfected with 0.5 μg of pGL4-miR-1199/2210011C24Rik or pGL4 Firefly luciferase promoter reporter by using Lipofectamine 3000 (Invitrogen). Additionally, cells were transfected with 10 ng of pRL-CMV (Promega), which encodes a Renilla luciferase and was used for subsequent cell number normalization.

To test miR-1199/2210011C24Rik promoter activity upon Zeb1 gain and loss of function studies, NMuMG/E9 cells were plated as described before and reverse transfected with 20 nM of siRNAs against Zeb1 or a negative control (Ambion). On the next day, cells were transfected with the different reporter constructs mentioned above and treated with TGFβ for 3 days. For Zeb1 gain of function studies, NMuMG/E9 cells were forward transfected with 50 ng of pCS3-6xMyc-tag or pCS3-6xMyc-tag-Zeb1 along with the different luciferase reporters.

*Zeb1 3′UTR reporter*: NMuMG/E9 cells were plated and reverse transfected with 20 nM of miR-1199-5p, miR-200b-3p, miR-429-3p pre-miR miRNA precursors or a negative control (Ambion) as described above. On the next day, cells were transfected with 0.2 μg of pMIR-Report Zeb1-3′ UTR/miR-1199/miR-200b/miR429 wild-type or mutant vector along with 10 ng of pRL-CMV by using Lipofectamine 3000 (Invitrogen).

**RNA isolation and quantitative RT-PCR.** In order to quantify gene transcripts, total RNA was isolated using Tri Reagent (Sigma-Aldrich) following the manufacturer's instructions. Complementary DNA (cDNA) was generated by reverse transcription of RNA using M-MLV reverse transcriptase (Promega) and was quantified by real-time PCR using Mesa Green qPCR MasterMix plus (Eurogentec). Riboprotein L19 was used as internal normalization control. FCs were calculated using the comparative Ct method (ΔΔCt). Primers used for quantitative RT-PCR are listed in Supplementary Table 4.

For quantitative miRNA analysis, total RNA was isolated using miRNeasy kit (Qiagen). Mature miRNAs were reverse-transcribed by using the miRCURY LNA Universal RT miRNA PCR kit (Exiqon). MiR-1199-5p (Exiqon; 206004) expression was measured by RT-PCR and normalized to U6 small nuclear RNA (Exiqon; 203907) expression as internal control. FCs in miRNA expression were calculated using the comparative Ct method (ΔΔCt).

**Immunoblotting.** Cells were lysed on ice in RIPA buffer (50 mM Tris-HCl (pH 8.0), 150 mM NaCl, 10% glycerol, 1% NP40, 0.5% NaDOC, 0.1% SDS, 2 mM $MgCl_2$, 2 mM $CaCl_2$, 1 mM DTT, 1 mM NaF, 2 mM $Na_3VO_4$ and 1× protease inhibitor cocktail (Sigma-Aldrich)) for 20 min, centrifuged and protein concentrations were determined via Bio-Rad Bradford solution according to the manufacturer's instructions. Proteins were mixed with SDS-PAGE loading buffer (10% glycerol, 2% SDS, 65 mM Tris, 1 mg/100 ml bromophenol blue, 1% β-mercaptoethanol) and equally loaded on a SDS polyacrylamide gel. After size fractionation, proteins were transferred on an Immobilon-P PVDF membrane (Millipore) using an electrophoretic transfer cell (Bio-Rad) and blocked with 5% skim milk powder dissolved in TBS/0.05% Tween for 1 h. The membrane was incubated with primary antibodies at 4 °C overnight or at room temperature for 1 h. HRP-conjugated secondary antibodies were used to visualize specific proteins on a Fusion Fx7 chemoluminescence reader (Vilber Lourmat). Uncropped immuno blot scans from main blots are displayed in Supplementary Fig. 8.

**Immunofluorescence microscopy.** Cells were grown on uncovered glass cover slips (Menzel–Glaser), fixed with 4% paraformaldehyde/PBS for 20 min, permeabilized with 0.5% NP40 for 5 min and blocked with 3% BSA/0.01% Triton X-100/PBS for 30 min. Afterwards, cells were incubated at room temperature with the primary antibodies (listed above) diluted in 3% BSA/0.01% Triton X-100/PBS for 1.5 h, followed by an incubation with a fluorophore-coupled secondary antibody (Alexa Fluor, Invitrogen) for 1 h. Cell nuclei were visualized with DAPI (Sigma-Aldrich). After staining, the cells were mounted in fluorescence mounting medium (Dako) on microscope slides and imaged using fluorescence microscopy (Leica DMI 4000).

**EMT phenotypic microscopy-based screen.** The general setup of the screen has been described recently[23]. In brief, NMuMG/E9 cells were plated as duplicates in a 96-well plate and reverse transfected with 20 nM pre-miR miRNA precursors (Ambion; Supplementary Table 4) using Lipofectamine RNAiMax (Invitrogen) 2 days prior to the addition of TGFβ for 4 days. Epithelial control cells were transfected for 3 days in the absence of TGFβ. For immunofluorescence analysis, cells were processed as described above and mesenchymal characteristics, such as formation of focal adhesions (paxillin), actin stress fibres (phalloidin) and fibronectin deposition, were stained as described in ref. [23]. Images of cells in a 96-well plate were taken with an Operetta HCS microscope (Perkin Elmers). EMT features were quantified (Columbus software, version 2.5.0) and the median of two replicates was calculated and compared to the median value of miR-Ctr-transfected cells. Standard deviations (s.d.) were estimated. MiRNAs showing at least 3× s.d. for all three EMT read-outs or 4× s.d. in two EMT read-outs aside from the median value of miR-Ctr-transfected cells were considered as hits to block or induce EMT. Some miRNAs, which turned out to be no hits in the EMT phenotypic microscopy-based screen, but maintained an epithelial cell morphology of NMuMG/E9 cells cultured in the presence of TGFβ (judged by eye), were also considered hits.

**Trans-well migration and invasion assays.** Boyden chamber invasion assay (24-well plate format): mesenchymal Py2T and 4T1 cells were reverse transfected with 50 nM of pre-miR miRNA precursors in a six-well plate. Two days later, the cells were trypsinized, washed once with PBS and counted. A total of 25,000 cells were plated as duplicates in a 24-trans-well migration or invasion insert (membrane with 8 μm pores, covered with a layer of growth factor reduced Matrigel; BD Biosciences). After 18 h, the cells were fixed with 4% paraformaldehyde/PBS, nuclei were stained with DAPI (Sigma-Aldrich) and non-migrated/invaded cells on the upper surface of the membrane were removed with a cotton swab. Migrated/invaded cells on the bottom of the membrane were imaged with a fluorescence microscope (Leica DMI 4000) and quantified. As chemo-attractant, a gradient of 0.2–20% FCS was used.

Boyden chamber migration screen (96-well plate format): mesenchymal Py2T cells were reverse transfected with 50 nM of pre-miR miRNA precursors in a 96-well plate. After 2 days, cells were trypsinized and washed with PBS in a round-bottom 96-well plate (Corning). MiR-Ctr-transfected cells were counted and 6000 cells were plated as duplicates in the inserts of a 96-well FluoroBlok fluorescent-blocking high-density positron emission tomography insert (BD Biosciences) and,

in parallel, in a 96-well reference plate (Cell Carrier 96, Perkin Elmers). The same cell suspension volume was used to plate cells transfected with other pre-miR miRNA precursors. As chemo-attractant, a gradient of 0.2–20% FCS was used. Migrated cells (bottom of membrane of migration insert) as well as cells in the reference plate were stained with DAPI and imaged with an Operetta HCS microscope (Perkin Elmers), quantified (Columbus software, version 2.5.0) and normalized to each other.

**Chromatin immunoprecipitation.** ChIP experiments were performed as previously described[44]. Briefly, crosslinked protein-bound DNA of Py2T cells was sonicated (Bioruptor, Diagenode) to achieve chromatin fragments of an average size of 300 bps. For ChIP of endogenous Zeb1, 150 μg of chromatin was incubated with 10 μg of Zeb1 antibody (sc-25388) and immunocomplexes were precipitated with 40 μl of pre-blocked magnetic Protein G beads (Invitrogen). Immunocomplexes were eluted from the beads, de-crosslinked, and genomic DNA was purified by phenol/chloroform extraction and precipitated with sodium acetate. One out of forty of the ChIP sample and 1% of input DNA were used for quantitative RT-PCR. Fold enrichments for specific miR-1199 promoter regions were calculated by IP over input samples and normalized to isotype-specific IgG as negative control. Primers targeting different genomic regions of the miR-1199 promoter are listed below:

−52/ + 51 bp from TSS: fwd: 5′-AGTTGTGCCCCTGTCTCG-3′; rev: 5′-GATGGGCGTCTCCACTCTG-3′.

−185/−87 bp from TSS: fwd: 5′-CCTCTGGAGAGGAGCACTTG-3′; rev: 5′-CCCCAGTACCTCCGTTATACT-3′.

−265/−376 bp from TSS: fwd: 5′-TATTTGGGCATCTCAATTCG-3′; rev: 5′-GACCCCAGGACTCCACTCTC-3′.

−604/ + 504 bp from TSS: fwd: 5′-CAGTCAACCCGTGTGTTCAG-3′; rev: 5′-GTCACCTGCCAGCTGGATT-3′.

**Microarray data and analysis.** In order to compare the expression of the miR-1199-5p in human cells, we downloaded raw data (CEL files) of the experiment GSE32474[25] from the Gene Expression Omnibus database. Data was analysed in R (https://www.r-project.org)/Rstudio (https://www.rstudio.com) using Bioconductor add-on packages. The background signal correction, normalization and summarization were performed by robust multiarray averaging[45] from the affy package[46]. To identify differences in gene expression between epithelial and mesenchymal samples, the linear models for microarray data (limma)[47] package was used.

**miRNA sequencing and analysis.** Total RNA was isolated from NMuMG/E9 cells undergoing TGFβ-induced EMT by using the miRNeasy Mini Kit (Qiagen) according to the manufacturer's instructions. Quality and quantity of total RNA was analysed by using the RNA 6000 Pico kit from Agilent. MiRNA-sequencing libraries of two biological replicates were prepared with the Illumina TruSeq Small RNA Sample Prep as described by the manufacturer. cDNA libraries were size fractionated on a 6% Tris-Borate-EDTA (TBE) gel and fragments of 140–160 nt were extracted. Libraries were loaded and sequenced on an Illumina HiSeq 2500 using protocols defined by the manufacturer.

For bioinformatics analysis, sequencing adapters were removed from obtained single-end miRNA-seq reads (51-mers) using the preprocessReads function from R/Bioconductor package QuasR, version 1.0.9 43: 'preprocessReads(filename = tab $FileName, outputFilename = tab$FileNameCleaned, Rpattern "TGGAATTCTC GGGTGCCAAGGAACTCCAGTCAC")'. Preprocessed reads were mapped to mouse genome assembly, version mm10, with Bowtie allowing up 50 hits in the genome (44, included in the QuasR package) using the command: 'qAlign ("samples_preprocessed.txt", "BSgenome.Hsapiens.UCSC.mm10", maxHits = 50)'. The coordinates of mature miRNAs downloaded from miRBase (http://www.mirbase.org/, v21) were extended by 3 bp on each side and passed as an argument to the qCount function, by which we quantified expression as the number of reads that started within any mature miRNA. The differentially expressed miRNAs were identified using the edgeR package (version 3.2.4). MiRNAs with FDR <0.05 and minimum log2 FC of ±2 were used for further analysis.

**Whole-transcriptome RNA sequencing and analysis.** NMuMG/E9 cells were transfected with pre-miR miRNA precursors (Ambion) as described before prior to 4 days TGFβ treatment or not. Total RNA was isolated from cells of two independent experiments using the RNeasy Mini Kit (Qiagen) according to the manufacturer's instruction. RNA quality control was performed with a fragment analyser using the standard or high-sensitivity RNA analysis kit (DNF-471-0500 or DNF-472-0500) from Labgene and RNA concentration was measured by using the Quanti-iT™ RiboGreen RNA assay Kit (Life Technologies/Thermo Fisher Scientific). A total of 200 ng of RNA was utilized for library preparation with the TruSeq stranded total RNA LT sample prep Kit (Illumina). Poly-A + RNA was sequenced with HiSeq SBS Kit v4 (Illumina) on an Illumina HiSeq 2500 using protocols defined by the manufacturer.

Obtained single-end RNA-seq reads (51-mers) were mapped to the mouse genome assembly, version mm10, with default RNA-STAR[48], parameters except for allowing only unique hits to genome (outFilterMultimapNmax = 1) and filtering reads without evidence in spliced junction table (outFilterType = "BySJout").

Using RefSeq mRNA coordinates from UCSC (http://www.genome.ucsc.edu, downloaded in December 2015) and the qCount function from QuasR package (version 3.12.1), we quantified gene expression as the number of reads that started within any annotated exon of a gene. The differentially expressed genes were identified using the edgeR package (version 1.10.1). Genes with FDR <0.05 and minimum log2 FC of ±1 were used for downstream analysis.

**Tumour transplantation**. A total of $0.5 \times 10^6$ 4T1 cells stably expressing ZsGreen-mmu-miR-1199/-200b/-429 or ZsGreen only were injected into the mammary fat pad of 7–9-week-old female NMRI nude or NSG mice. Mice were killed after 18 or 21 days post injection, respectively, and lungs were isolated. Tissue processing and haematoxylin and eosin staining were performed as previously described[24]. Lung sections were imaged using an Axio Imager scanning microscope (Zeiss), and the number of lung metastasis as well as the area per lung nodule were quantified. Mann–Whitney $U$ test was used for statistical analysis. All studies involving mice were approved by the Swiss Federal Veterinary Office (SFVO) and the regulations of the Cantonal Veterinary Office of Basel Stadt (licence 1907).

**Isolation of circulating tumour cells**. Eighty microlitres of blood was drawn by heart puncture of tumour-bearing mice killed 21 days post orthotopic cell injection, mixed with normal growth medium and cultured in a six-well plate for 5 days. Adherent cells were washed with PBS daily. A MTT (3-(4,5-dimethylthiazol-2-yl)-2-5-diphenyltetrazolium bromide) staining was used to visualize tumour cell colonies. In brief, cells were incubated with 0.4 mg/ml MTT solution diluted in normal growth medium for 3 h. Afterwards, cell colonies were imaged and quantified.

**Statistical analysis**. Statistical analyses and graphs were generated using GraphPad Prism software (version 6). Statistical analyses were performed using an unpaired, two-sided $t$ test or Mann–Whitney $U$ test with $*P < 0.05$, $**P < 0.01$, $***P < 0.001$, $****P < 0.0001$.

**Data availability**. The miRNA expression data of a TGFβ time course experiment in NMuMG/E9 cells as well as RNA expression data of NMuMG/E9 cells ectopically transfected with miR-Ctr, miR-1199-5p, miR-200b-3p or miR-429-3p are deposited at Gene Expression Omnibus (GEO, accession number: GSE86026). All other remaining data are available within the article and supplementary files, or available from the authors upon request.

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

## Acknowledgements

We thank T. Brabletz for sharing vector constructs with us. We are grateful to R. Goosen for bioinformatics support. Furthermore, we thank A. Vettiger, P. Schmidt, H. Antoniadis and I. Galm for technical assistance. We are grateful to T. Roloff, S. Dessus-Babus, C. Beisel and the Genomics Facility Basel for next-generation RNA sequencing. Calculations were performed at sciCORE (http://scicore.unibas.ch/) scientific computing core facility at University of Basel. This work was supported by the SystemsX.ch RTD project Cellplasticity, the SystemsX.ch MTD project MetastasiX, the Swiss National Science Foundation and the Swiss Cancer League. N.M.-S. was supported by a Marie-Heim Vögtlin grant from the Swiss National Foundation.

## Author contributions

M.D. designed and performed the experiments, analysed the data and wrote the paper. S.T., F.L., N.M.-S. and M.S. designed and performed experiments and analysed the data. R.I. and R.K.R.K. performed bioinformatics analyses. G.C. oversaw the project, designed experiments, analysed data and wrote the paper.

## Additional information

**Competing interests:** The authors declare no competing financial interests.

