## [Peer review file · Nature Communications]

Reviewers' comments:

Reviewer #1 (Remarks to the Author):

Diepenbruck et al. describe a new addition to the microRNAs that impede EMT, miR-1199. They show that miR-1199, like members of the miR-200 family, is induced in mouse cells by TGFb, is capable of inhibiting ZEB1 and in turn is transcriptionally inhibited by ZEB1, making a double negative feedback loop. MiR-1199 is not related to miR-200 in sequence and has few targets in common, but like miR-200 it inhibits cell migration and invasion in vitro, and metastasis in vivo, consistent with its targeting of ZEB1. The data are of good quality and their description and interpretation are clear.

Major comments

1. A surprising aspect that is not commented on and needs to be commented on by the authors is the fact that miR-1199 in humans has a different sequence from the murine miR-1199, including a difference in the seed region, which might be expected to endow the human miR-1199 with different targeting properties. The authors show that miR-1199 can inhibit EMT in human as well as mouse cells, but was it the mouse miRNA that was overexpressed in the human cells? The binding site in human ZEB1 is the same sequence as in the mouse, so it may not be efficiently targeted by human miR-1199.
2. It is unfortunate that the authors were not able to inhibit the endogenous miR-1199 to verify that the levels induced by TGFb are sufficient to have the effects on ZEB1 and EMT properties that the overexpressed miRNA clearly has. Demonstrating this would strengthen the report.
3. I see in miRBase that the hairpin structure of the human miR-1199 pre-miR is atypical and also that there are very few sequence reads associated with mature miR-1199, raising some concern that its expression in humans may be very low. The authors should discuss this.

Minor comments

4. Line 180. I did not immediately understand what is meant by "log2FC(+/-1)". The authors may want to spell this out the first time it is used.
5. Line 235. This would read better as "Because miR-1199-5p regulates the expression of Zeb1, we assessed..."
6. Fig. S3 – a subfigure could be included to show the effectiveness of the ZEB1 knockdown.
7. Fig. 5b – It is interesting that there is no effect of myc-Zeb1 in the absence of TGFb. Perhaps this could be commented on.
8. Fig S3b – it would be interesting to see whether mutating the Zeb1 binding site at -18 in the promoter of the Zeb-luc reporter abrogates induction by TGFb.
9. It is surprising that miR-200b and miR-429 have such different effects on metastasis (Fig. 6) given they have identical seed sequences and a large overlap in genes regulated (Fig. 7b). Perhaps the authors could comment on this.
10. Figs 3, 7. Venn diagrams are more informative when the areas are in proportion to the numbers.

Reviewer #2 (Remarks to the Author):

In this manuscript, the authors identified a novel miRNA (miR-1199-5p) regulating the process of EMT in an animal model of breast cancer.

Particularly, miR-1199-5p was found to be a new component of the reciprocal double-negative feedback back loop including miR-200 family and ZEB1 regulating the acquisition of EMT phenotype, migration/invasion ability and metastatic potential.

The authors performed a time course miR-sequencing analysis evaluating the changes of miRNA during the TGF-b-induced EMT process in normal murine mammary gland cells.

This represents an interesting approach to evaluate miRNA level changes during the EMT process.

The conclusions are convincing and original. Furthermore, it is interesting the analysis on the commons and distinct target of the miR-200 family and miR-1199-5p. This represents an interesting approach to dissect the levels of individual contribution of these miRNAs.

Some concerns and issues should be addressed on the screening process that identified miR-1199-5p.

In fig. 1a are reported the 32 miRNAs that are differentially expressed after 96h of TGF- β -induced EMT. Then, to identify the function of these miRNAs, functional study was performed using miRNA mimic transfection.

It is not clear why miRNA mimics and TGF- β stimulation were used for those miRNAs whose levels were already increased by TGF- β . Indeed, their levels are already upregulated after 96h of TGF- β stimulation. To identify their functions, miRNA inhibitors should be used.

It makes sense to use miRNA mimic alone (without TGF- β stimulation) to evaluate their possible effect on EMT for those miRNA that are upregulated after 96h of TGF- β stimulation. Indeed, miR-145a-5p (Supp. Fig. 1 b) induced decreased levels of E-Cadherin, according to IF microscopy.

It would be useful to add histogram graphs on the levels of E-Cadherin (quantification of average expression levels) to better evaluate the differences in the expression levels across the different miRNA mimic transfection.

In Fig.1 a and in Supp. Fig. 2, it is reported miR-504-5p among the downregulated miRNA, but in the text and in the Supp. fig. 1b it is reported miR-504-3p.

In fig. 5 b, the ** to indicate statistical significant differences are missing.

Statistical analysis was conducted appropriately.

Manuscript: NCOMMS-17-00762-T

miR-1199-5p and Zeb1: a novel double-negative feedback loop coordinating EMT and tumour metastasis

Maren Diepenbruck, Stefanie Tiede, Meera Saxena, Robert Ivanek, Ravi Kiran Reddy Kalathur, Fabiana Lüönd, Nathalie Meyer-Schaller, and Gerhard Christofori

To be seen by the reviewers

Introductory remarks

We appreciate the reviewers' comments and constructive criticisms on our manuscript. We have used the past four months to address all the points raised by the reviewers. In total, we have performed a large number of additional experiments to further characterize the function of miR-1199-5p during an EMT.

In particular, we have spent major efforts to study the role of miR-1199-5p by additional loss of function experiments. Towards this aim, we have used commercial miRNA inhibitors and we have generated miRNA sponges to interfere with the endogenous levels of miR-1199-5p and miR200b/c-3p. Even though these various strategies have worked on a technical level as shown by control experiments with 3'UTR reporter constructs, we did not observe any effects on Zeb1 expression nor on the EMT process itself by interfering with endogenous miR-1199 or miR-200 functions. The details of these results are presented in Point 2 of the responses to Reviewer #1. From the results we conclude that the transcriptional and posttranscriptional regulation of the key EMT TF Zeb1 by miR-200 family and by miR-1199 is very complex. We suppose that taking out one miRNA (miR-1199-5p or miR-200b/c-3p) is not sufficient to affect Zeb1 expression or "incline" the epithelial system, at least in the model systems we have employed.

Moreover, we have examined miR-1199-5p expression in human breast cancer cell lines and have further studied the activity of the miR-1199 promoter. In particular, we have tested the different E-box motifs within the promoter region for Zeb1 binding. Furthermore, we have characterized the other EMT-associated miRNAs identified in our screen by additional epithelial and mesenchymal EMT marker expression.

In addition, a large number of minor additional experiments and changes in Figures and Text have been performed as detailed in the point-by-point reply below.

The results from the additional experiments complement and validate the results of the previous version of the manuscript. Altogether, the report is now providing new insights into the function and regulation of a new miRNA, miR-1199-5p, during an EMT *in vitro* and primary tumour growth and metastasis formation *in vivo*. The data also expand the complexity of the reciprocal miR-200 – Zeb1 regulation by adding another reciprocal repression of Zeb1 with miR-1199.

As a consequence of the major revisions, the text, some of the Figures and, in particular, the presentation and interpretation of the results have been revised to accommodate the new data. New panels have been added to the original Fig. 5 and to Supplementary Fig. 1, 3 and 4.

Point-by-point reply

Reviewer #1

Diepenbruck et al. describe a new addition to the microRNAs that impede EMT, miR-1199. They show that miR-1199, like members of the miR-200 family, is induced in mouse cells by TGF β , is capable of inhibiting ZEB1 and in turn is transcriptionally inhibited by ZEB1, making a double negative feedback loop. MiR-1199 is not related to miR-200 in sequence and has few targets in common, but like miR-200 it inhibits cell migration and invasion in vitro, and metastasis in vivo, consistent with its targeting of ZEB1. The data are of good quality and their description and interpretation are clear.

We appreciate the reviewers' positive and constructive comments. In the following point-by-point reply we have addressed all major and minor technical aspects pointed out by the reviewer to improve the quality and scientific impact of our work.

1. A surprising aspect that is not commented on and needs to be commented on by the authors is the fact that miR-1199 in humans has a different sequence from the murine miR-1199, including a difference in the seed region, which might be expected to endow the human miR-1199 with different targeting properties. The authors show that miR-1199 can inhibit EMT in human as well as mouse cells, but was it the mouse miRNA that was overexpressed in the human cells? The binding site in human ZEB1 is the same sequence as in the mouse, so it may not be efficiently targeted by human miR-1199.

The reviewer's concern about an efficient downregulation of Zeb1 in MCF10A cells by the human miR-1199-5p is clearly of importance, since the mouse and the human miR-1199-5p seed match sequence differ in one base, however, their seed sequence in the 3'UTR of Zeb1 is the same.

In our studies in MCF10A cells we have utilized a human miR-1199-5p mimic construct and we apologize that it has not been clear from our study description. We now marked the use of hsa-miR-1199-5p mimic in the text as well as in the figure legends (Supplementary Fig. 3 and 4).

Although the seed sequence in the 3'UTR of Zeb1 and the seed match in the hsa-miR-1199-5p are not 100% complementary, forced expression of this miRNA in MCF10A cells cultured in the absence and presence of TGF β led to a significant reduction in Zeb1 mRNA (Supplementary Fig. 3f) and protein levels (Supplementary Fig. 4h). These results imply that even if a seed sequence of a target RNA and a seed/match sequence of a miRNA are not perfectly complementary, a target gene can still be efficiently downregulated in its expression at the posttranscriptional level. This adds a certain degree of flexibility and, consequently, more complexity to miRNA target gene determination. These considerations have now been added in the text.

2. It is unfortunate that the authors were not able to inhibit the endogenous miR-1199 to verify that the levels induced by TGF β are sufficient to have the effects on ZEB1 and EMT properties that the overexpressed miRNA clearly has. Demonstrating this would strengthen the report.

We completely agree with the reviewer that loss-of-function studies for miR-1199-5p would help to shed additional light on the function of miR-1199-5p in EMT and breast cancer progression.

Therefore, we once more attempted two different loss-of-function strategies (Figure to be seen by the reviewers: Reviewer Fig. R1). First, we generated different miRNA sponge constructs for miR-1199-5p, miR-200b/c-3p and a scrambled construct as negative control. These constructs consisted of five bulged binding sites for either miR-1199-5p or miR-200b/c-3p or five scrambled sites (not recognized by any murine miRNA) inserted in the 3'UTR of an eGFP reporter gene driven by the CMV promoter. We transiently transfected NMuMG/E9 cells with these miRNA sponges and observed a downregulation in expression of GFP in the cells co-transfected with miR-1199-5p or miR-200b-3p mimics. This reduction in GFP

expression was not observed in cells transfected with the scrambled miRNA sponge or in cells co-transfected with a miR-Ctr mimic and the miR-1199-5p or miR-200b/c-3p sponge construct (Reviewer Fig. R1a). Additional Zeb1 3'UTR luciferase reporter assays confirmed the functionality of the miRNA sponge constructs (Reviewer Fig. R1b,c). Here, forced expression of miR-1199-5p and miR-200b-3p mimics in NMuMG/E9 cells significantly increased the Zeb1 3'UTR reporter activity in cells co-transfected with miR-1199-5p or miR-200b/c-3p sponge construct compared to cells transfected with the scrambled sponge construct. However, miRNA sponge-mediated interference with endogenously expressed miR-1199-5p as well as miR-200b/c-3p led only to a small increase in Zeb1 3'UTR reporter activities in epithelial NMuMG/E9 cells (Reviewer Fig. R1b,c).

Next, we assessed whether the loss of miR-1199-5p or miR-200b/c-3p in NMuMG/E9 cells would affect the expression levels of the direct target gene Zeb1 or would induce or advance the EMT process. Transient transfection of the miRNA sponge constructs in NMuMG/E9 cells cultured in the absence and presence of TGF β did not lead to altered cell morphology in comparison to cells transfected with a scrambled sponge construct. Additionally, mRNA as well as protein levels for Zeb1 and other EMT markers were unaffected (Reviewer Fig. R1d-f).

As second strategy for miRNA loss-of-function studies we utilized miRNA inhibitors purchased from two different companies (Ambion and Exiqon). We transiently transfected NMuMG/E9 cells in the absence or presence of TGF β with the miRNA inhibitors and tested their functionality using the Zeb1 3'UTR luciferase reporter described above. Similar to the results obtained with miRNA sponges, the miR-1199-5p inhibitors led only to a small increase in Zeb1 3'UTR reporter activity compared to anti miR-Ctr transfected cells (Reviewer Fig. R1g). Furthermore, no increase in Zeb1 expression and no difference in EMT marker expression were observed at the mRNA (Reviewer Fig. R1f) and protein (Reviewer Fig. R1i) level.

From these results we conclude that both strategies to inhibit miRNA function in NMuMG/E9 cells have worked at a technical level. However, the regulation of Zeb1 expression at the transcriptional and posttranscriptional level seems more complex, and we suppose that the loss of one Zeb1 regulator (miR-1199-5p or miR-200b/c-3p) is not sufficient to alter Zeb1 cellular levels. These regulatory networks, especially the overlapping double-negative feedback loops between miRNAs and TFs, may act as a "buffered system" to compensate for the loss of one regulator and ensure Zeb1's functionality. Taking out more than one of the Zeb1 regulators at the same time would be the next step to unbalance the overlapping double-negative

feedback loops.

Another thought that comes to mind is that that an EMT induced by TGF β relies on the repression of miR-200 and miR-1199 expression by Zeb1. Zeb1 is upregulated in its expression by TGF β -signaling, for example, by the activity of Sox4 (Tiwari et al., 2013), and thus EMT is “jumpstarted” by strong upstream regulators. In contrast, repression of miR-200 family members or miR-1199 by individually may not suffice to “jumpstart” the system. The expression and activation of other factors induced by Sox4 and/or the Zeb, Snail and Twist family transcription factors may be required.

Reference

Tiwari N, Tiwari VK, Waldmeier L, Balwierz PJ, Arnold P, Pachkov M, Meyer-Schaller N, Schübeler D, van Nimwegen E, Christofori G. (2013) Sox4 is a master regulator of epithelial-mesenchymal transition by controlling Ezh2 expression and epigenetic reprogramming. *Cancer Cell* 23, 768-783.

3. I see in miRBase that the hairpin structure of the human miR-1199 pre-miR is atypical and also that there are very few sequence reads associated with mature miR-1199, raising some concern that its expression in humans may be very low. The authors should discuss this.

To address the reviewers' concern about miR-1199 expression in humans, we have analysed its expression in different human breast cancer cell lines using the dataset GSE32474 available on the Gene Expression Omnibus. We analysed the expression of E-cadherin and Zeb1 in these breast cancer cell lines as well and observed that high/low levels of E-cadherin correlate with high/low expression of miR-1199, respectively. In contrast, Zeb1 and miR-1199 display an anti-correlative expression profile in MCF7, BT-549, HS578T and MDA-MB-231 cells (Supplementary Fig. 3c).

4. Line 180. I did not immediately understand what is meant by “log2FC(+/-1)”. The authors may want to spell this out the first time it is used.

We apologize for the confusion. We now spell out this abbreviation for the first time we use it in the text.

5. Line 235. This would read better as “Because miR-1199-5p regulates the

expression of Zeb1, we assessed...

We appreciated the reviewers' suggestion and changed the sentence in the manuscript accordingly.

6. Fig. S3 – a subFig. could be included to show the effectiveness of the ZEB1 knockdown.

In Supplementary Fig. 4d-e we have now added the missing siRNA-mediated knockdown efficiencies at the RNA and protein levels for Py2T, 4T1, MCF10A and NMuMG/E9 cells.

7. Fig. 5b – It is interesting that there is no effect of myc-Zeb1 in the absence of TGF β . Perhaps this could be commented on.

We observed that the transient transfection efficiency of the 6xMyc-tag-Zeb1 or 6xMyc-tag control constructs in NMuMG/E9 cells was very low, i.e. only few cells have been transfected. We suggest this is the reason why we observed no (in epithelial cells) or only a trend (4d TGF β) in miR-1199-5p repression by analysing the whole cell populations consisting of few transfected and mostly untransfected cells (Fig. 5b). In the miR-1199 luciferase promoter assays shown in Fig. 5a, we co-transfected the 6xMyc-tag-Zeb1 or 6xMyc-tag control vector along with the luciferase reporter construct, which are most likely taken up together by one cell. Therefore, a strong and significant transcriptional repression by Zeb1 has been observed in these experiments.

To overcome the technical problem with low transfection efficiencies, we have attempted to generate stable Zeb1-overexpressing NMuMG/E9 cells to measure miR-1199-5p expression by quantitative RT-PCR analysis. However, after retroviral infection of NMuMG/E9 cells with a 6xMyc-tag-Zeb1-IRES-GFP construct, we were repeatedly unable to detect Zeb1 overexpression by immunoblotting analysis (Reviewer Fig. R1j). Interestingly, after infection of the 6xMyc-tag-Zeb1-IRES-GFP construct, flow cytometry-sorted GFP⁺ cells remained GFP⁺, yet they did not express Zeb1. These observations may indicate that NMuMG/E9 cells do not tolerate high expression levels of Zeb1, a hypothesis that needs to be experimentally tested.

8. Fig S3b – it would be interesting to see whether mutating the Zeb1 binding site at -

18 in the promoter of the Zeb-luc reporter abrogates induction by TGF β .

In response to the reviewers' suggestion, we generated four different miR-1199 luciferase reporter constructs from the original wild type reporter used in Fig. 5a. Each reporter exhibit one mutated Zeb1 E-box binding motif (CAGGTG to CATTG) as illustrated in Fig. 5c. We transiently transfected each of these individual miR-1199 *Firefly* luciferase promoter constructs along with a *Renilla* luciferase reporter and a 6xMyc-tag-Zeb1 or the 6xMyc-tag control construct in epithelial NMuMG/E9 cells. As shown in the new Fig. 5e, cells transfected with the E-box 1 (-18 bps from TSS) mutant reporter displayed a partial rescue in luciferase activity upon forced expression of Zeb1 compared to the wild type or the other three promoter mutant reporters.

From these results we conclude that the E-box 1 binding site within the miR-1199 promoter is of importance for Zeb1 binding and transcriptional target gene repression. However, it seems not be the only regulatory motif since the E-box 1 mutant reporter is not able to match the luciferase activity levels observed in 6xMyc-tag control-transfected cells.

9. It is surprising that miR-200b and miR-429 have such different effects on metastasis (Fig. 6) given they have identical seed sequences and a large overlap in genes regulated (Fig. 7b). Perhaps the authors could comment on this.

The reviewer is completely right; the different effects of miR-200b-3p and miR-429-3p on lung metastasis formation were also surprising to us.

MiR-200b-3p and miR-429-3p have a lot of genes in common during EMT, which can be explained by an identical seed sequence as pointed out by the reviewer. However, each miRNA has also its own pool of (direct) regulated genes during EMT (Fig. 7b,d), therefore different phenotypic outputs in the context of cancer metastasis are in principle possible. In need of appropriately responding to the reviewer, we have also contacted experts in the field and, notably, Prof. Greg Goodall (Centre for Cancer Biology, SA Pathology and University of South Australia, Adelaide, Australia) has communicated to us that his laboratory has also not observed any discernible effect of the forced expression of miR-200b on metastasis of 4T1 cells *in vivo* (unpublished data; personal communication). The mechanistic basis for this observation remains elusive, however, we now mention these observations and the personal communication in the text to let the readers know.

10. Figs 3, 7. Venn diagrams are more informative when the areas are in proportion to the numbers.

We have followed the reviewers' suggestion and remodelled the Venn diagrams in Fig. 7b,d.

Reviewer #2

In this manuscript, the authors identified a novel miRNA (miR-1199-5p) regulating the process of EMT in an animal model of breast cancer.

Particularly, miR-1199-5p was found to be a new component of the reciprocal double-negative feedback back loop including miR-200 family and ZEB1 regulating the acquisition of EMT phenotype, migration/invasion ability and metastatic potential. The authors performed a time course miR-sequencing analysis evaluating the changes of miRNA during the TGF- β -induced EMT process in normal murine mammary gland cells.

This represents an interesting approach to evaluate miRNA level changes during the EMT process.

The conclusions are convincing and original. Furthermore, it is interesting the analysis on the commons and distinct target of the miR-200 family and miR-1199-5p. This represents an interesting approach to dissect the levels of individual contribution of these miRNAs.

We thank the reviewer for the positive and constructive comments on our work.

1. In fig. 1a are reported the 32 miRNAs that are differentially expressed after 96h of TGF- β -induced EMT. Then, to identify the function of these miRNAs, functional study was performed using miRNA mimic transfection.

It is not clear why miRNA mimics and TGF- β stimulation were used for those miRNAs whose levels were already increased by TGF- β . Indeed, their levels are already upregulated after 96h of TGF- β stimulation. To identify their functions, miRNA inhibitors should be used.

The reviewer is right that we should have used a loss-of-function approach for those miRNAs, which are transcriptionally downregulated during TGF β -induced EMT of NMuMG/E9 cells, in particular to ask the question whether a miRNA is able to block TGF β -induced EMT? Yet as we now discuss in the revised manuscript and also in

point 2 of our responses to reviewer #1, we had difficulties in using miRNA inhibitors for a repression of an EMT, therefore, we utilized miRNA mimics for functional screening of the 32 EMT-regulated miRNAs. In the group of transcriptionally downregulated miRNAs, we asked the question, whether a miRNA is sufficient to induce an EMT in the absence of TGF β or to advance an EMT in the presence of TGF β upon miRNA mimic transfections. Using this approach we were able to identify miR-145a-3p and miR-6944-3p as partial EMT inducers. Transfection of these miRNAs in epithelial NMuMG/E9 cells induced a strong downregulation of E-cadherin and an upregulation of N-cadherin (Supplementary Fig. 1b,c).

It makes sense to use miRNA mimic alone (without TGF- β stimulation) to evaluate their possible effect on EMT for those miRNA that are upregulated after 96h of TGF- β stimulation. Indeed, miR-145a-5p (Supp. Fig. 1 b) induced decreased levels of E-Cadherin, according to IF microscopy.

It would be useful to add histogram graphs on the levels of E-Cadherin (quantification of average expression levels) to better evaluate the differences in the expression levels across the different miRNA mimic transfection.

To obtain a more comprehensive insight and strengthen our previous results, we followed the reviewer's suggestion and analysed the mRNA expression of the EMT markers E-cadherin, N-cadherin, fibronectin and Zeb1 in NMuMG/E9 cells (either untreated or treated for with TGF β for 4 days) for all transfections with miRNA mimics (Supplementary Fig. S1c).

In Fig.1 a and in Supp. Fig. 2, it is reported miR-504-5p among the downregulated miRNA, but in the text and in the Supp. fig. 1b it is reported miR-504-3p.

The reviewer is correct. We have corrected the mistake in the text as well as in Supplementary Fig. S1b.

*In fig. 5 b, the ** to indicate statistical significant differences are missing.*

We recalculated the statistical significance using an unpaired, two-sided t-test, but did not observe a significant difference in the graph shown in Fig. 5b.

Statistical analysis was conducted appropriately.

Figure to be seen by Reviewers

For the reviewers
Reviewer Figure R1

Additional results to be seen by the reviewers

Fig. Reviewer R1

(a-c) miR1199-5p and miR-200b/c-3p sponge constructs are functional.

(a) Downregulation of eGFP by miR-1199-5p or miR-200b-3p mimic co-transfected with pEGFP-miR-1199-5p or pEGFP-miR-200b/c-3p sponge constructs. Five miR-1199-5p, miR-200b/c-3p or scrambled bulged binding sites were inserted in the 3'UTR of an eGFP reporter gene driven by the CMV promoter. NMuMG/E9 cells were transiently transfected with a miR-1199-5p, a miR-200b-3p or a miR-Ctr mimic along with the indicated miRNA sponge constructs. (b,c) NMuMG/E9 cells described in (a) were further transfected with a Zeb1 3'UTR *Firefly* luciferase reporter construct exhibiting one miR-1199-5p seed sequence (b) or two miR-200b/429-3p seed sequences (c). Cells were co-transfected with a Renilla luciferase reporter and fold changes +/- s.e.m. of relative luminescence (Firefly/Renilla luminescence) were calculated (n=3).

(d-i) Loss of miR-1199-5p or miR-200b/c-3p is not sufficient to induce an EMT. (d-e) NMuMG/E9 cells were transiently transfected with the miRNA sponge constructs indicated. (d) Bright field images of NMuMG/E9 cells cultured in the absence and presence (2 days) of TGF β . Quantitative RT-PCR (e) and immunoblotting (f) analysis was performed for the expression of different EMT markers in NMuMG/E9 cells cultured as described before.

(g-i) NMuMG/E9 cells were transiently transfected with a miR-Ctr, miR-1199-5p or a miR-200c-3p inhibitor from two different companies (Ambion and Exiqon). (g) A Zeb1 3'UTR *Firefly* luciferase reporter with one miR-1199-5p binding site was used as described in (b) (n=1). Quantitative RT-PCR (h) and immunoblotting (i) analyses was performed for the expression of different EMT markers in NMuMG/E9 cells transfected with the indicated miRNA inhibitors from Ambion and cultured in the absence and presence (2 days) of TGF β .

(j) Failure to establish NMuMG/E9 cells stably expressing high levels of Zeb1. NMuMG/E9 cells were infected with the retroviral vector pMys-6xMyc-tag-Zeb1-IRES-GFP or with the negative control vector pMys-6xMyc-tag-IRES-GFP. GFP positive cells were sorted by flow cytometry and Zeb1 protein levels were analysed by immunoblotting analysis of two independent infection events (n=2). GAPDH was used as loading control.

REVIEWERS' COMMENTS:

Reviewer #1 (Remarks to the Author):

The authors have adequately addressed my comments, notwithstanding the lack of success of their attempts to verify that inhibition of the endogenous miR-1199-5p suppresses the proposed effect of the miRNA. This is discussed in the revised manuscript and a possible explanation provided. Furthermore, the regulatory interconnectedness between miR-1199-5p and ZEB1 help support the likelihood of the major conclusions. Nevertheless, without the final proof provided by inhibition of the endogenous miR-1199-5p, it remains a possibility that the microRNA exerts an effect only at the higher levels achieved by overexpression. Consequently I suggest the title be changed to the less definitive "miR-1199-5p and Zeb1: a novel double-negative feedback loop potentially coordinating EMT and tumour metastasis"

Minor correction:

Line 359 "surrprising"

Reviewer #2 (Remarks to the Author):

This is an improved revision of an interesting manuscript.

Also, in the rebuttal, the authors described well what were the challenges and how they solved these.

A manuscript of interest for the large spectrum of the readers of the journal.

Manuscript: NCOMMS-17-00762-T

miR-1199-5p and Zeb1 function in a double-negative feedback loop potentially coordinating EMT and tumour metastasis

Maren Diepenbruck, Stefanie Tiede, Meera Saxena, Robert Ivanek, Ravi Kiran Reddy Kalathur, Fabiana Lüönd, Nathalie Meyer-Schaller, and Gerhard Christofori

To be seen by the reviewers

Introductory remarks

We appreciate the positive comments of the reviewers on our revised manuscript and addressed all their remaining concerns/comments in a second revision. Additionally, we have adapted the format of our manuscript to comply the journals' format requirements.

Reviewer #1 (Remarks to the Author):

The authors have adequately addressed my comments, notwithstanding the lack of success of their attempts to verify that inhibition of the endogenous miR-1199-5p suppresses the proposed effect of the miRNA. This is discussed in the revised manuscript and a possible explanation provided. Furthermore, the regulatory interconnectedness between miR-1199-5p and ZEB1 help support the likelihood of the major conclusions. Nevertheless, without the final proof provided by inhibition of the endogenous miR-1199-5p, it remains a possibility that the microRNA exerts an effect only at the higher levels achieved by overexpression. Consequently I suggest the title be changed to the less definitive "miR-1199-5p and Zeb1: a novel double-negative feedback loop potentially coordinating EMT and tumour metastasis"

The reviewer is right. We have changed the title of the manuscript ("miR-1199-5p and Zeb1 function in a double-negative feedback loop potentially coordinating EMT and tumour metastasis") as suggested by the reviewer.

Minor correction:

Line 359 "surrprising"

We have corrected the mistake in the text.

Reviewer #2 (Remarks to the Author):

This is an improved revision of an interesting manuscript. Also, in the rebuttal, the authors described well what were the challenges and how they solved these. A manuscript of interest for the large spectrum of the readers of the journal.

We thank the reviewer for his positive feedback on our revised manuscript.